# Sedimentary ancient DNA reveals a threat of warming-induced alpine habitat loss to Tibetan Plateau plant diversity

Sisi Liu[1,2], Stefan Kruse [1], Dirk Scherler [3,4], Richard H. Ree [5], Heike H. Zimmermann[1], Kathleen R. Stoof-Leichsenring [1], Laura S. Epp[1], Steffen Mischke [6] & Ulrike Herzschuh [1,2,7 ✉]

Studies along elevational gradients worldwide usually find the highest plant taxa richness in mid-elevation forest belts. Hence, an increase in upper elevation diversity is expected in the course of warming-related treeline rise. Here, we use a time-series approach to infer past taxa richness from sedimentary ancient DNA from the south-eastern Tibetan Plateau over the last ~18,000 years. We find the highest total plant taxa richness during the cool phase after glacier retreat when the area contained extensive and diverse alpine habitats (14–10 ka); followed by a decline when forests expanded during the warm early- to mid-Holocene (10–3.6 ka). Livestock grazing since 3.6 ka promoted plant taxa richness only weakly. Based on these inferred dependencies, our simulation yields a substantive decrease in plant taxa richness in response to warming-related alpine habitat loss over the next centuries. Accordingly, efforts of Tibetan biodiversity conservation should include conclusions from palaeoecological evidence.

[1] Alfred Wegener Institute Helmholtz Centre for Polar and Marine Research, Polar Terrestrial Environmental Systems, Potsdam, Germany. [2] Institute of Environmental Science and Geography, University of Potsdam, Potsdam, Germany. [3] GFZ German Research Centre for Geosciences, Potsdam, Germany. [4] Institute of Geological Sciences, Freie Universität Berlin, Berlin, Germany. [5] Negaunee Integrative Research Center, Department of Science and Education, Field Museum, Chicago, USA. [6] Institute of Earth Sciences, University of Iceland, Reykjavík, Iceland. [7] Institute of Biochemistry and Biology, University of Potsdam, Potsdam, Germany. ✉email: Ulrike.Herzschuh@awi.de

Global change affects mountain biodiversity and alters ecosystem functioning, eventually threatening the provision of ecosystem services to human society[1]. Whether ongoing glacier retreat, treeline rise, and land-use intensification will result in an increase or decrease of high-elevation plant richness is heavily debated[2–6]. Studies of elevational transects locate the highest plant richness at intermediate elevations, which are mostly found within the forest belt in temperate regions. For example, plant diversity peaks at ~3,600 m a.s.l[7]. (metres above sea level) on the south-eastern Tibetan Plateau (Hengduan Mountains, Fig. 1a, red dotted line) which harbours one third of the vascular plant flora of China[8]. However, it is unclear whether this widely observed hump-shaped diversity pattern represents a sampling effect (because mid-elevations are often preferentially sampled)[9], an area effect (because mid-elevation areas are mostly overrepresented and thus can support higher plant diversity)[9], a mid-domain effect (because species distributions overlap mainly at the geometric centre even without environmental gradients)[10], or whether it is really the mild temperatures and/or ecotone effect of forests that support a high richness[9]. Hence, the question remains as to whether plant diversity at high elevations will increase with temperature-driven treeline advance, as would be expected when simply projecting an upward movement of vegetation belts.

Alpine plants, which typically include many endemic taxa, are threatened by habitat loss when treelines rise[11] and are therefore a focus of conservation considerations. It is uncertain whether their preferred habitats[12] or a diversity of habitats should be conserved to protect richness[13]. It is also unclear whether landscape diversity resulting from extensive and traditional land use can, to some extent, compensate for climate-change effects on alpine plant richness[14].

These uncertainties mainly originate from a lack of long-term biodiversity records. Most projected mountain diversity changes are based on knowledge obtained from samples taken across a spatial extent and include various artefacts[15]. The advantages of a time-series approach over the traditional space-for-time approach are that the sampled site is constant (i.e., normalizing for the sampling effect), that sampling elevation always represents the same portion of the investigated mountain range (i.e., normalizing for the area effect) and is always placed at the same relative

elevation (i.e., normalizing for the mid-domain effect). Hence, such an approach can well reflect the temporal biodiversity-environment relationship and as such increase the effects of relevant variables when predicting biodiversity change over time[16]. Mountain lake sediments are historical archives of ecological change, but classical vegetation proxies, such as pollen or macrofossils, are not suitable indicators for plant diversity change[17]. However, methodological advances in sedimentary ancient DNA (sedaDNA) metabarcoding now allow an assessment of biodiversity at higher taxonomic resolution than traditional approaches through time[18].

Here, we reconstruct the taxonomic richness of plants in the catchment area of Lake Naleng (Hengduan Mountains, south-eastern margin of the Tibetan Plateau; Fig. 1a) over the past ~18,000 years. We apply a sedaDNA metabarcoding approach with general plant primers (Methods) to 72 horizons from a lake sediment core. We consider temperature, habitable area, forest shifts, and human impact as potential drivers of changes in plant taxa richness. In particular, we investigate whether richness increases with forest invasion into alpine habitats, recapitulating the modern elevational richness gradient. Finally, we make inferences for plant taxa richness under future climate change by applying the inferred long-term diversity-environmental relationships and provide suggestions for a future plant diversity conservation planning. We find that the total plant taxa richness is highest during late glacial times, when the area was dominated by alpine meadows, and lowest during the early Holocene when forest extent was at its maximum. By analogy to the past, we infer that total plant-taxa richness could decrease in the future due to a warming-induced upward movement of the treeline.

## Results and discussion

**Plant DNA and taxa richness changes.** After bioinformatic filtering of raw sequencing output (Methods), 6,021,603 sequence counts were obtained from 138 PCR (polymerase chain reaction) replicates of 71 investigated sediment horizons. They were assigned to 218 terrestrial seed plant taxa with 100% best identity (Supplementary Data 1). A few PCR replicates, mainly from the late-glacial period, had to be excluded from further analyses because they were without read counts (Supplementary Fig. 1a).

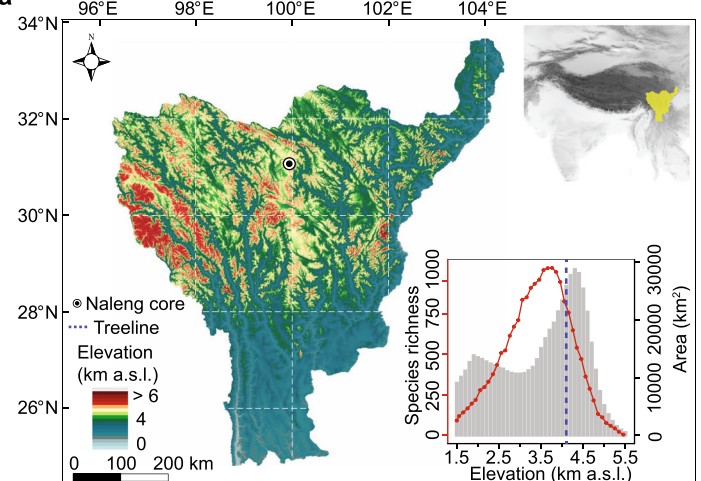

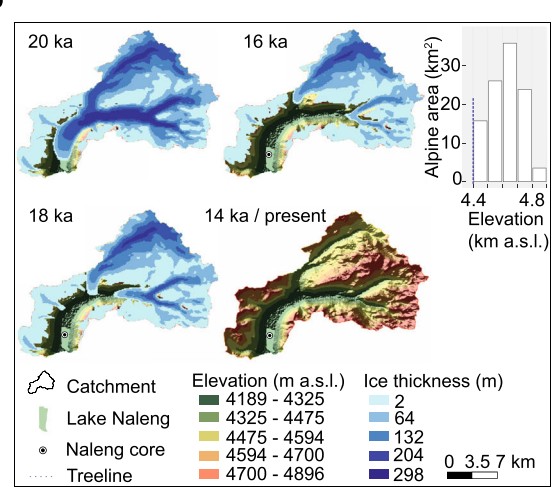

**Fig. 1 The Naleng lacustrine core was recovered from the centre of Lake Naleng (black bullet point), in the Hengduan Mountains, a designated biodiversity hotspot in East Asia. a** Location of the Hengduan Mountains on the south-eastern Tibetan Plateau, China (top-right inset, yellow fill). Area-elevation relationship (grey bars), elevational species richness distribution[7] (red dotted line), and forest zone (blue dashed line) are shown in the lower-right inset. **b** Lake Naleng catchment area is 128 km². The simulation of the glacier extent (Methods) indicates that the Lake Naleng catchment became ice-free by about 14 ka. The extent of alpine area per 100-m elevation is shown as white bars.

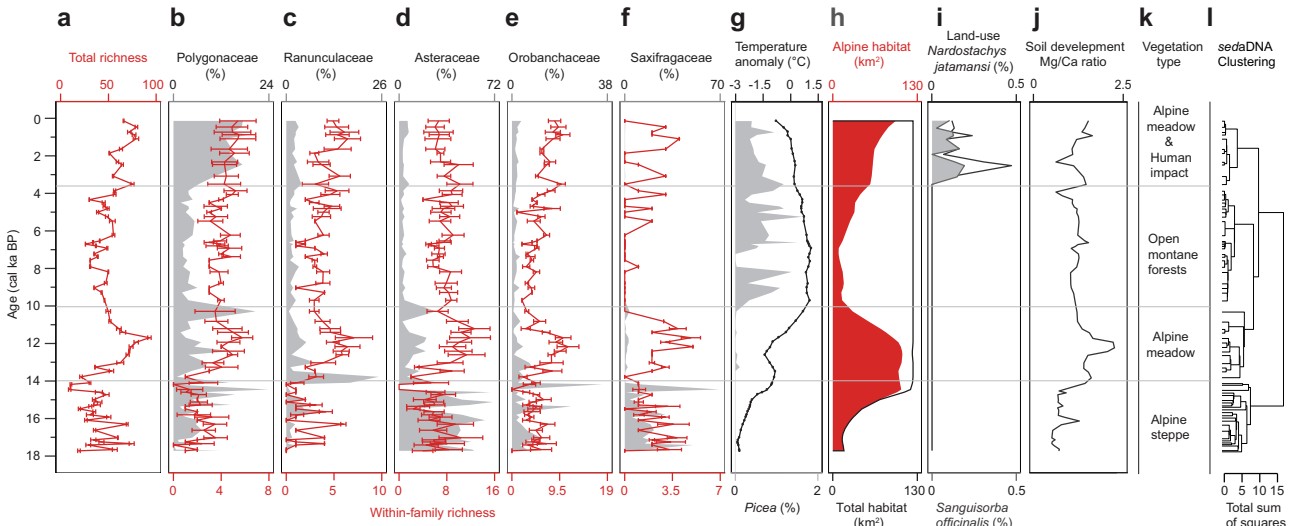

**Fig. 2 Long-term history of plant sedaDNA recorded in Lake Naleng compared with time-series data. a** Total plant richness (red line; n = 138 PCR replicates of 71 lake-sediments, bars indicate 95% confidence intervals). **b–f** Comparison of sedimentary ancient DNA abundance (in %, grey polygon) of the most common alpine plant families with corresponding within-family plant richness (red line; n = 138 PCR replicates of 71 lake-sediments, bars indicate 95% confidence intervals) for Polygonaceae, Ranunculaceae, Asteraceae, Orobanchaceae, and Saxifragaceae. **g** The Northern Hemisphere (30°–90°N) temperature anomaly record since last deglaciation based on multiple proxies[57,58] (black line with points, Methods) and percentage of *Picea* sedaDNA (grey polygon). **h** Alpine habitat area (red polygon) within the Lake Naleng catchment is the sum of pixels above the treeline (4400 m a.s.l.) based on simulated total habitat area (black outline, Methods). **i** sedaDNA indicators of traditional land-use including *Sanguisorba officinalis* (grey polygon) and *Nardostachys jatamansi* (black line). **j** The Mg/Ca ratio of Lake Naleng indicates the soil development within the lake catchment[29]. **k** vegetation types inferred from the pollen record[19,20] and sedaDNA record. **l** Zonation (horizonal grey lines) according to a stratigraphically constrained cluster analysis (CONISS) based on relative read abundance. Data are presented as mean ± 95% confidence interval (error bars) in a–f. Source data are provided with this paper.

Apart from 5 outliers with high read numbers, PCR replicates yielded read counts of a similar order of magnitude (Supplementary Fig. 1b). We found no correlation between read counts and total plant taxa richness (df = 69, rho = 0.014, p = 0.908). From this we conclude that read count has no impact on the inferred diversity signal. Also, neither variations in plant taxa richness nor compositional signals differed between results from single PCR samples or from pooled-PCR samples (Supplementary Fig. 1c, d, Supplementary Table 1 and Supplementary Table 2). Accordingly, we assume that the variations of plant taxa richness over time can be reliably tracked by pooling results from PCR replicates of one horizon.

Overall the sedaDNA record reproduces the compositional vegetation changes (Fig. 2k) inferred from pollen data[19,20] (Supplementary Fig. 2) and pollen-based vegetation change agrees with other pollen records from the Tibetan Plateau:[21,22] alpine steppe dominated 18–14 ka, alpine meadow 14–10 ka, open *Picea* forest 10–3.6 ka and alpine meadow after 3.6 ka with the presence of typical land-use indicators such as *Sanguisorba officinalis* (Fig. 2i, grey area)[20] and *Nardostachys jatamansi* (a traditional Tibetan medicinal plant[23]) (Fig. 2i, black line). The sedaDNA better captures the vegetation signals within the lake catchment than pollen as it is not impacted by upward plant material transport (Supplementary Fig. 2) and records more taxa at higher taxonomic resolution than pollen spectra (Supplementary Table 3). Accordingly, and because the lake catchment covers the most common elevations in the Hengduan Mountains (~4200 to ~4900 m a.s.l., Fig. 1a, b), we conclude that Lake Naleng archived the main signal of the south-eastern Tibetan alpine ecosystem. This reasoning aligns with a modern study that indicates a non-random vegetation composition in the alpine belt of the Hengduan Mountains and identified phylogenetic clustering of alpine plant taxa in connection with environmental filtering[24].

Total plant taxa richness was relatively low before 14 ka, higher between 14 and 10 ka, low again between 10 and 3.6 ka and high after 3.6 ka (Fig. 2a). Similar trends were obtained for taxa richness within important alpine plant families (Fig. 2b–f). Proportional immigration of taxa dominated during 14–10 ka and after 3.6 ka compared to their respective previous time period (Supplementary Fig. 3, Methods). Considering the restricted entries in the taxonomic database (EMBL Nucleotide Database standard sequence release 127[25], we redid the sedaDNA analyses using a 95% best identity threshold for taxa assignment which yielded 984 unique sequence types (Supplementary Data 2) indicating the possibility of additional plant taxa. Plant taxa richness patterns based on the 95% best identity are similar to those with 100% best identity (Supplementary Fig. 4), which provides confidence in our results. Furthermore, a similar temporal pattern of total plant taxa richness was obtained when analysing the data before taxonomic assignment and the data containing all terrestrial seed plant sequences (Methods), as indicated by the highly significant correlations of these time-series with total plant taxa richness (Supplementary Table 1). Sample processing-related errors (e.g., PCR and sequencing) may have slightly inflated taxa richness. However, we assume that we rather underestimate taxa richness because of non-specificity of the marker and non-completeness of the reference database, which likely means that the sedaDNA detected taxa number is lower than the absolute taxa number recorded in the flora. Additionally, species-rich families[26] in the flora including Asteraceae, Saxifragaceae, and Orobanchaceae have highest richness in our record. Thus, plant taxa richness (total plant taxa richness and taxa richness within dominant alpine families) can be regarded as a semi-quantitative proxy of taxa richness in our study.

Taken together, our applied plant sedaDNA metabarcoding identifies more plant taxa at lower taxonomic level than any other

palaeo-approach before, providing a first reliable record of relative plant taxa richness variation on a millennial timescale for the Tibetan Plateau.

**Drivers of plant taxa richness changes**. Total plant taxa richness shows contrasting correlations with multiple proxy-based temperature reconstructions (Fig. 2g, dotted line) for different periods of the record: a positive correlation for 18–10 ka (rho = 0.225) but negative correlations for 14–3.6 ka (rho = −0.728) and 10–0 ka (rho = −0.932; Table 1). This suggests that temperature is unlikely to be a direct driver of plant taxa richness but may instead trigger different environmental processes that lead to contrasting biodiversity-temperature correlations.

Our analyses reveal a weak positive but statistically non-significant correlation between total plant taxa richness and total habitat area (Fig. 2h, black line) in response to glacier retreat at the end of the last glacial in the catchment of Lake Naleng (rho = 0.257, alpha level = 0.25; Table 1). Thus, we find no evidence to support the idea that total plant taxa richness mainly depends on the available area. Underestimation of this dependency might be related to limitations in our glacier-extent modelling approach that uses a proxy-based temperature reconstruction averaged from the Northern Hemisphere rather than from our study area. Nevertheless, to our knowledge, our study is the first palaeo time-series approach that addresses the extensively debated relationship between taxa richness and area[27]. Aside from habitable area, further processes related to rapid glacier retreat negatively impacted total plant taxa richness changes during the late glacial (rho = −0.587, alpha level = 0.025; Table 1). This might be attributed to disturbances on unstable slopes restricting vegetation establishment[28]. The increase in pedogenic minerals (as indicated by sedimentary proxy Mg/Ca ratio from the same record[29], Fig. 2j) may have promoted the increase of richness of some alpine families (e.g. Polygonaceae, alpha level = 0.025; Ranunculaceae, alpha level = 0.025; Orobanchaceae, alpha level = 0.05, Supplementary Table 4), supporting the idea that soil development contributes to the coexistence of a large number of plant species[30]. However, it is not the key driver for the total plant taxa richness (alpha level = 0.25, Table 1).

We find a strong positive relationship between total plant taxa richness and the alpine habitat extent (Fig. 2h, red area; which itself negatively correlates with sedaDNA signals of *Picea*) in the catchment area after 14 ka (alpha level < 0.05; Table 1). Hence, in contrast to our expectation from the modern elevational plant taxa richness gradients in the Hengduan Mountains that peaks in the upper forest belt (Fig. 1a), early Holocene forest expansion into the catchment of Lake Naleng did not result in a plant taxa richness increase but in a richness decrease (alpha level = 0.01; Table 1). Accordingly, the reconstructed and simulated late-Holocene forest retreat (i.e., alpine area extent) also correlates with a richness increase (alpha level = 0.0005, Table 1). We assume that the retreat of forests is related to late Holocene cooling and weakening of the Asian summer monsoon, not to human impact, and is supported by a lack of late Holocene forest burning[20].

Interestingly, sedaDNA results show that relative abundance and plant taxa richness within high-alpine plant families such as Asteraceae, Orobanchaceae, and Saxifragaceae can differ substantially, such that we find high within-family richness but low relative abundance at 14–10 ka (Fig. 2d–f). Of course, the relationship between relative read abundance and relative abundance of the taxon in the vegetation is still poorly understood and previous studies indicate that biases originate from, for example, PCR setup (e.g. preference for short reads and

**Table 1 Summary of correlation coefficients between total plant taxa richness and the predictor variables.**

| | 18-10 ka | | | | | 14-3.6 ka | | | | | 10-0 ka | | | | |
|---|---|---|---|---|---|---|---|---|---|---|---|---|---|---|---|
| | rho | adj p-value | df | adj df | alpha level | rho | adj p-value | df | adj df | alpha level | rho | adj p-value | df | adj df | alpha level |
| Total habitat | 0.257 | 0.524 | 34 | 10 | 0.25 | — | — | — | | | — | — | — | | |
| Temperature | 0.225 | 0.746 | 34 | 11 | >0.25 | −0.728 | 1.32e-06 | 35 | 11 | 0.01 | −0.932 | 2.00e-15 | 33 | 11 | 0.0005 |
| Glacier's decay | −0.587 | 0.002 | 30 | 13 | 0.025 | — | — | — | | | — | — | — | | |
| Alpine habitat | — | — | — | | | 0.739 | 7.24e-07 | 35 | 11 | 0.01 | 0.966 | 3.18e-20 | 33 | 11 | 0.0005 |
| Forested area | — | — | — | | | −0.739 | 7.24e-07 | 35 | 11 | 0.01 | 0.966 | 3.18e-20 | 33 | 11 | 0.0005 |
| Mg/Ca ratio | 0.381 | 0.088 | 34 | 11 | 0.25 | 0.412 | 0.046 | 35 | 11 | .25 | 0.164 | 1.000 | 33 | 11 | >0.25 |
| Land-use | — | — | — | | | — | — | — | | | 0.939 | 3.87e-16 | 33 | 11 | 0.0005 |

rho: Spearman's Rank correlation coefficient
adjusted p-value: two-tailed with Bonferroni adjustment
df: degrees of freedom
adjusted df: adjusted degrees of freedom
alpha level: directional alpha levels of critical values for Spearman's Rank correlation coefficient
/not a predictor variable in corresponding time transition
predictor variable with alpha level ≤0.05 is in bold

reads with high GC content[31]). However, studies of modern lake sediments have also shown that the compositional differences among sites are preserved[32,33]. Similarity in compositional changes between Lake Naleng sedaDNA and the pollen record supports this finding (Supplementary Fig. 2). We speculate that simultaneous high diversity of alpine habitats and maximum alpine habitat extent during 14–10 ka in the catchment of Lake Naleng may have provided habitats for many different plant taxa thereby suppressing domination by a few taxa[34], which could, in turn, have created microenvironments that facilitate novel taxa migration.

We assume that the signs for late Holocene grazing intensification in sedaDNA and pollen records (Supplementary Fig. 2) are related to human impact. They indicate that human impact started or substantially increased after 3.6 ka, which aligns with archaeological evidence[35]. Their positive correlation with total plant taxa richness in the palaeorecord (10–0 ka, rho = 0.939, alpha level = 0.0005, Table 1) is consistent with the findings from experimental studies that moderate land use can increase taxa richness[14]. However, we find that the positive effect is smaller compared to the negative effect of alpine habitat loss due to forest invasion (10–0 ka, rho = 0.966, alpha level = 0.0005, Table 1).

**Potential pattern of future plant taxa richness**. In summary, we identified alpine habitat extent as the best predictor variable for total plant taxa richness (explained deviance = 96.04%, Supplementary Table 5), which was then used in a generalized linear model (Fig. 3a). The simulations project extensive alpine habitat loss in the Lake Naleng catchment area over the next 250 years in response to a predicted 2.5 °C warming (Supplementary Fig. 5),

leading to a pronounced decrease in total plant taxa richness (Fig. 3b) and restricting cold-adapted taxa to high-mountain regions (Fig. 3c). In particular, taxa of endemic-rich high-alpine families are likely to disappear from the catchment (Supplementary Fig. 6). Because similar habitats will be rare in the surrounding Hengduan Mountains and the amplification of warming in high-elevation central Tibetan areas[36] is unfavourable for these taxa, these taxa may become extinct. An upward expansion of montane taxa and a loss of high alpine taxa in the study area agree with predictions from a comprehensive species distribution modelling approach for the Hengduan Mountains[4].

Our approach has several shortcomings. It assumes that treeline change is sensitive to temperature change in the region. Although this assumption is supported by palaeoecological evidence showing that forests expanded into higher elevations under warming during the early- to mid-Holocene and retreated to lower elevations during the late Holocene cooling[22], and by modern observations of an upward shifting treeline on the southeastern Tibetan Plateau in the past 100 years[37], the pace of treeline response is observed to lag the temperature warming in some mountain regions due to a variety of processes. Such processes include interspecific competition, forest-shrub interactions, dispersal variations, or even extreme climate events[37,38]. So, our temperature-treeline-richness relationship may therefore be correct on a millennial time scale but may overestimate changes on shorter time scales. Furthermore, our approach considers plant richness as a whole or focus on certain alpine families. Therefore, the habitat gain and loss of individual taxa or specific functional groups cannot be evaluated. Hence, our simulated taxa loss in relation to shrinking alpine habitat extent should be treated as a potential pattern by analogy to the past. It requires

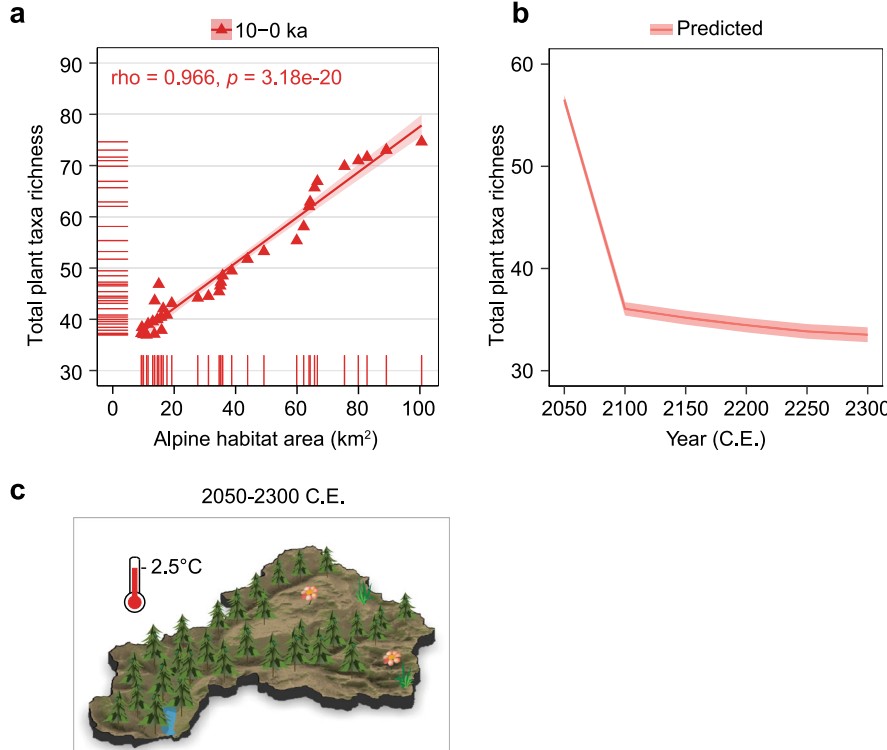

**Fig. 3 Predicted total plant taxa richness under 2.5 °C climate warming between 2050 and 2300 based on the inferred past relationship between total plant taxa richness and alpine habitat area. a** The relationship between total plant richness and alpine habitat area was established by a generalized linear model (Methods). **b** Prediction of total plant richness loss based on predicted alpine area habitat loss as inferred from simulating the forest rise in response to warming. **c** Visualization of total plant richness in Lake Naleng's catchment after 2.5 °C warming. Data are presented as mean ± 95% confidence interval (red area) in **a** and **b**.

**Fig. 4 Visualization of total plant taxa richness and effects of abiotic factors on plant richness across four time-intervals.** We calculated the statistical relationship between total plant richness and predictor variables (rounded rectangle) between consecutive periods of time (Methods). Alpha significance codes of Spearman correlation are ***0.0005, **0.01, and *0.025 according to adjusted degrees of freedom. The mean annual temperature anomaly is indicated by the thermometer. Positive and negative correlation is marked in red and blue font, respectively. The catchment sketches illustrate that disturbance in the glaciated landscape was likely of importance during the deglaciation period. Our results indicate that once the catchment became ice-free, alpine habitat extent is the main driver of total plant richness while land-use is only of secondary importance during the late Holocene. From the switch in correlation sign, we assume that temperature is likely not a direct driver of richness change.

confirmation from a more sophisticated species-specific approach that also considers realistic migrational lags.

**Implications for conservation of future plant taxa richness.** With respect to conservation efforts of the unique Hengduan Mountains diversity hotspot the following conclusions can be derived from our time-series approach (Fig. 4). First, to maintain high total plant richness, nature conservation should focus on alpine habitats. This contrasts with the conclusions based on elevational gradients which suggest prioritizing forests[39]. Focusing on alpine areas will also help to protect richness within alpine plant families that contain many endemic taxa as a result of Quaternary glaciations and geographical isolation[40]. Second, areas harbouring extensive alpine habitat and habitat diversity in the current upper alpine zone should be protected to provide space for warming-related upward plant migration. Third, any potential positive effects from grazing management are probably too weak to compensate for climate-change impacts on plant diversity in alpine habitats. Our study indicates that time-series investigations from palaeoecological investigations using sedaDNA can inform decision-making in nature conservation by revealing potential plant responses to changing environments and, when used alongside modelling studies of modern species distributions[41], create a fuller picture of plant dynamics.

## Methods

**Study site.** Lake Naleng (31.10° N, 99.75° E, 4200 m a.s.l.) is fed mainly via a major river channel on the northern side of the lake and drains from the southern margin. Several small streams from the adjacent mountains (up to 4,900 m a.s.l.) drain directly into the lake[29]. The basin of the lake was formed by glacial activity during the Last Glacial Maximum as indicated by erratic boulders and moraines[29]. The study area is influenced by the Indian summer monsoon, generating warm and wet conditions. Based on the instrumental data collected from Ganze station (31.62° N, 100.00°E; 3,522 m a.s.l.), the closest meteorological station about 80 km north-east of the lake, mean July temperature (MJT) is 14.3 °C and mean January temperature is −3.9 °C. Annual precipitation is about 620 mm with most falling from May to October. Yak and sheep livestock graze in the catchment during the summer. The vegetation composition has sharp environmental gradients:[19,20] (1) montane forests, consisting of conifers (*Abies aquamata*, *A. faxonia*, *Picea likiangensis*, *P. purpurea*), are found up to 4,400 m a.s.l. and distributed mainly on north-facing slopes; (2) broadleaved plants (*Betula*, *Rhododendron*) form a secondary canopy; (3) alpine meadow (e.g. *Polygonum*, *Kobresia*) is found above the subalpine eco-tone; (4) the high-alpine zone (4,900–5,200 m a.s.l.) is dominated by cushion and rosette plants (e.g. *Saussurea*).

**Material.** In total, 72 sediment samples were collected from the core. In the climate chamber at a temperature of −10 °C, about 2 mm of the exposed sediment of frozen samples was removed with a small single-use clean blade and the inner part used for ancient DNA isolation.

**Dating and chronology.** Dating and an age-depth model are described in detail in a previous publication[42]. As macrofossils were absent throughout the core, sixteen samples of bulk organic carbon were selected for accelerator mass spectrometer (AMS) [14]C dating at the Leibniz Institute Kiel. The determined lake reservoir effect of 1500 years was subtracted from each [14]C date, prior to calibration to calendar years (cal yr BP) using CALIB 5.0.1[43,44].

**DNA extraction, amplification, and high throughput next-generation sequencing.** All DNA work was carried out in ancient DNA dedicated facilities at Alfred Wegener Institute, Helmholtz Centre for Polar and Marine Research, using strict ancient DNA precautions and protocols. Each extraction batch included nine samples (3–10 g sample⁻¹) and one extraction control, which was treated with a partially modified protocol of PowerMax® Soil DNA Isolation kit (Mo Bio Laboratories, Inc. USA). The isolation of DNA was first processed by loading 15 mL PowerBead solution, 1.2 mL C1 buffer, 0.8 mg proteinase K (VWR International), 0.5 mL 1 M dithiothreitol (VWR International), and samples into PowerBead tubes. Then, all tubes were vortexed in 10 min and incubated at 56 °C in a rocking shaker overnight under the aluminium foil protection. The subsequent extraction steps followed the manufacturer's instructions of the kit and were completed on the second day and DNA was eluted in 1.6 ml C6 buffer. For amplification we used plant universal primers g and h targeting the P6 loop of the chloroplast *trnL* (UAA) intron (Supplementary Table 6)[45]. To distinguish the samples after sequencing, both primers were modified by adding an 8 bp tag with at least five different base pairs between each to the 5′ end[46] and three additional NNNs for improving cluster detection on Illumina sequencing platforms[47]. Altogether, 25 μL per PCR reaction were prepared with the following reagents: Primers (forward: 5′ NNN(8 bp tag)GGGCAATCCTGAGCCAA 3′, reverse: 5′ NNN(8 bp tag)CCATTGAGTCTCTGCACCTATC 3′) with the final concentration of 0.4 μM, 1× Platinum® Taq DNA Polymerase High Fidelity PCR buffer (Invitrogen, USA), 0.25 mM dNTPS, 0.8 mg Bovine Serum Albumin, 2 mM MgSO₄ (Invitrogen, USA), 1 U Platinum® Taq High Fidelity DNA Polymerase (Invitrogen, USA) and 3 μL of sedaDNA template. PCRs (polymerase chain reactions) were run in the Post-PCR area separate from the ancient DNA facilities at 94 °C for 5 min (initial denaturation), followed by 50 cycles of 94 °C for 30 s, 50 °C for 30 s, 68 °C for 30 s and a final extension at 72 °C for 10 min. A no template control (NTC) was included for each PCR batch which included nine DNA extractions and one extraction control. PCR set-ups were conducted under a dedicated UV working station in the detached ancient DNA laboratory physically separated from the workplace of Post-PCR where we did the thermal cycling, purification, and pooling. Each PCR batch was replicated until obtaining two positive PCR replicates for each lake-sediment sample when the associated controls were negative. A qualified positive PCR product was considered only if it matched two conditions: (1) the gene band is evidently longer than that of negative controls; (2) the brightest staining is in the 100–200 bp range. Specifically, the thin/blurry products below 50 bp in corresponding controls are primer dimers and not expected PCR products. PCR products were visualized with 2% agarose gel electrophoresis, purified with the MinElute PCR Purification Kit (Qiagen, Germany), and measured with the ds-DNA BR Assay and the Qubit® 2.0 fluorometer (Invitrogen, USA) using 1 μL of PCR product. All purified PCR products were equimolarly pooled and sent away for sequencing to *Fasteris* SA, which used the MetaFast library protocol prior to sequencing on an Illumina HiSeq 2500 sequencing platform with paired-end reads of 125 bp length with the mode HiSeq High Output Version 4 by applying the HiSeq SBS Kit v4. Our project was sequenced together with another unknown sequencing project on a full HiSeq 2500 lane and resulted in 9.5 Gb with 37,922,797 generated clusters ≥Q30.

**Sequence analysis and taxonomic assignment.** The sequence data were analysed using OBITools software[48]. First, the paired-end DNA sequence reads were aligned using the *illuminapairedend* program in order to assemble the forward and reverse sequence reads. With the program *ngsfilter* paired sequences were assigned to samples based on their unique tag combination used for each sample. Afterwards read counts were summarized for unique sequence types using *obiuniq*, and *obigrep* was used to discard those sequences with a length <10 bp and a total count <10 reads in whole dataset. Subsequently, *obiclean* was used to exclude sequence variants which are likely attributed to PCR or sequencing errors by determining the sequences into head, internal, and singletons based on sequence count and similarity within one sample. Finally, sequence types were taxonomically assigned using the *ecotag* program, which was run on two reference databases: EMBL Nucleotide Database (standard sequence release 127[25]) and Arctic and Boreal vascular plant and bryophyte reference libraries[49–51]. The applied EMBL database was created by using an in silico PCR[52] with the g/h primers allowing five mismatches between primer and the targeted sequences of the EMBL entries to increase the taxonomic breadth.

**sedaDNA data quality control.** To further denoise, only those sequence types that were assigned to terrestrial seed plants and have a best identity value greater than or equal to 0.95 were kept for the following data processing. Sequence counts <10 in each sample were replaced with 0 using R software[53]. Subsequently, we excluded assumed contaminations from fruits, cultivars and taxa not occurring in China, which were Musaceae, PACMAD clade, Lycopersicon, BOP clade, Maleae, etc. from the dataset. The extraction and PCR blanks were mostly without any contamination, only in a single NTC (sample ID: NTC6, Supplementary Data 1) were some plant DNA fragments detected. However, we did not remove these sequences from its controlled samples (Supplementary Discussion). In addition, sequences were only considered as genuine if the corresponding taxa could be found in the study area (Hengduan Mountains), which was aided by an international open access database[54]. We collected all sequence types with the best identity value of 1 and a frequency ≥2, into a dataset named bestid1, while the bestid0.95 dataset included the sequences with best identity ≥0.95 and frequency ≥2. Sample ESL024 was excluded from further statistical analysis because no reads were obtained (Supplementary Data 1 and Data 2).

**Glacier-extant and habitat area simulation.** Several previous studies have shown that the climate at a millennial time scale on the eastern Tibetan Plateau is strongly impacted by monsoons, particularly the East Asian summer monsoon, which tracks changes in the westerlies and continental warming that are largely a function of mid- to high-latitude changes[55,56]. Thus, the past climate change in our study was inferred by the synthesized record of Northern Hemisphere (30°–90° N) temperature anomaly since the last deglaciation[57,58]. The physical surface area is a prerequisite for suitable habitat development and taxa shifts. Hence, we modelled the past glacier-cover changes first (Fig. 1b). We considered the catchment area that is available for plant colonization to be ice-free. Past ice extents were estimated with the numerical ice-flow model GC2D[59]. We ran simulations on the present-day topography, based on a 90-m resolution SRTM digital elevation model. Climate was imposed through a vertical mass-balance profile that we estimated from present-day conditions. Based on the spatially averaged mean elevation of present-day glaciers in the vicinity[60], we estimated an equilibrium line altitude (ELA) of ~5200 m[61]. We estimated the maximum ice accumulation rate to be 0.25 m yr$^{-1}$, based on different gridded precipitation data sets (HAR[62], GPCC[63]). Guided by observations from modern Tibetan glaciers[64], and by matching the present-day distribution of ice cover in the wider region of our study area, we estimated a mass balance gradient of 0.0115 m yr$^{-1}$ m$^{-1}$. Glacier and snow cover through time were interpolated from the corresponding ΔELA based on an integrated temperature lapse rate of 0.55 °C 100 m$^{-1}$ in the Hengduan Mountains[65] and reconstructed past temperature. To clarify, the ice-flow model only simulated the past flow of ice and any concurrent advance and retreat.

We then modelled the available habitable area within the lake catchment back-in-time (Supplementary Fig. 7) according to the following steps: (1) delineate the catchment using the global 1-arcsecond (90-m) SRTM digital elevation model and downscale to 30-m resolution for simulation; (2) combine the two reconstructed past temperature records;[57,58] (3) calculate the relative elevation of each pixel based on the integrated temperature lapse rate of 0.55 °C 100 m$^{-1}$ the same as for the glacier model[65] for the catchment over the past 18 ka for each step of 500 years by the relative temperature change from the constructed temperature series in (2); (4) calculate the elevational range of the catchment over the past 18 ka under the effect of simulated glacier cover; (5) group the elevation values per 100-m elevational band ranging from 1000 m a.s.l. to 6000 m a.s.l. under the effect of simulated glacier cover and sum the pixels in all elevational bands as total habitable area (Fig. 2h, black line); (6) sum up the number of pixels above the modern treeline (~ 4,400 m a.s.l.) to obtain the alpine habitat area (Fig. 2h, red area); and (7) compute the elevational range above the modern treeline in the catchment (2050–2300 CE) under the projections of temperature according to RCP 4.5 emissions scenario (source: http://svn.zmaw.de/svn/cosmos/branches/releases/mpi-esm-cmip5/src/mod) for indicating the loss of alpine habitat under the ongoing climate warming (Supplementary Fig. 5). The modern treeline was calculated based on 49 current treeline points taken from high-resolution satellite images in

GoogleEarth™ and open-source data (Supplementary Fig. 8 and Supplementary Table 7). It should be noted that uncertainties in the simulation may arise from species interactions and potentially lagging treeline response to climate warming[37].

**Statistical analyses.** All statistical calculations were carried out using the R software[53]. All correlations were computed using the corr.test(method = "spearman") in the psych package[66]. The Spearman Rank Correlation (rho), non-adjusted/adjusted probability values (*p*-value/adjusted *p*-value) and sample size (n) were obtained. Both *p*-values are helpful to check the significance of correlation under the unadjusted degrees of freedom, which is equal to sample size (n) minus 2.

As plant taxa richness increases with read counts, we rarefied the data bestid1 to equal counts for each sample based on the minimal total read count occurring in the entire sedaDNA dataset (base count = 11,949) 100 times (Supplementary Code 1), as well as the data bestid0.95 (base count = 13,344).

To investigate potential methodological biases of the sedaDNA-based plant taxa richness, two additional datasets were set up: (1) metabarcoding data before taxonomic assignment (hereafter referred as non-ecotag); (2) all terrestrial seed plant sequences without further sequence filtering (hereafter referred as terSeq data). Both datasets were rarefied to their respective minimal total read count (16,209; 14,645) 100 times. We investigated whether plant taxa richness is correlated to read counts for both datasets and whether it is correlated with total plant taxa richness of bestid1 dataset using corr.test(adjust = "none").

To test if the plant taxa richness and composition are stable, we first collected the deeply sequenced PCR product for each lake sediment sample from the dataset bestid1 (hereafter referred as single data). Then, we rarefied these data 100 times based on the minimal read count (6,339) across all samples and calculated the plant taxa richness. The correlation between plant taxa richness from data bestid1 and single data was calculated using corr.test(adjust = "none"). Finally, Procrustes, and Protest analyses were applied to check whether sample scores and taxa scores of the first two PCA (principal component analysis) axes of the single data match those of the bestid1 dataset. Only those sequences with a maximum relative read abundance of 0.25% at least were kept. Double-square root transformation was applied before PCA analysis[67]. The procrustes(), protest() and rda(scale = FALSE) are available in the vegan package[68].

In order to calculate the plant taxa richness for each single taxonomic family, we divided sedaDNA sequence data into subsets of taxonomic families and rarefied these subsets to a cut-off value of 100 total read counts to minimize the effect of relative abundance of taxonomic family. Such richness signals of families with low read counts (e.g., Ranunculaceae) could be compared with richness signals of families having higher read counts (e.g., Asteraceae). Furthermore, four distinct vegetation zones were classified using the chclust(method = "CONISS") in vegan package[68] based on the rarefied relative read abundance. We summed up the samples in each zone and computed the mean value of read counts per zone. Then, we rarefied the zonal data to its minimal total read count (40,377) 100 times. Afterwards, we computed the total vegetation turnover (beta diversity) using the turnover() function in the codyn package[69] based on the rarefied zonal data (Supplementary Code 2).

To identify the main drivers of plant taxa richness, we calculated the correlation coefficient between plant taxa richness and driver variables using corr.test (adjust = "Bonferroni") (Supplementary Code 3 and Supplementary Data 3). We separated the complete richness time-series into three time-intervals each consisting of two consecutive vegetation zones (according to CONISS), i.e., 18–10 ka, 14–3.6 ka, 10–0 ka. This approach accounts for variation of driver importance throughout the record and can even reveal sign changes in the relationship between driver variables and plant taxa richness. We used the smoothed data for plant richness and two predictor variables (Mg/Ca ratio and land-use) in the processing of correlation calculation, so that we adjusted the degrees of freedom to get the effective independent variables of the smoothed data. We used gam() for land-use data smoothing as it included a large number of zeros and loess(span = 0.5) to smooth the Mg/Ca ratio and plant taxa richness. Consequently, to obtain the alpha level we compared rho values with exact critical values of Spearman's rho according to adjusted degrees of freedom. Only an alpha level ≤ 0.05 is considered statistically significant. The results of correlation are summarized in Table 1 and Supplementary Table 4. The related code is available in Supplementary Code 4. A generalized linear model (GLM) was built using glm (family = "gaussian") for total plant taxa richness and those families that are significantly related to the predictor variables (alpine habitat area and land-use indicator) during 10–0 ka (Supplementary Code 5). This period was selected as it covers the warmest and most modern phase of the record. Moreover, the correlation between total plant taxa richness and alpine habitat is highest in this time interval. The temporal resolution of the correlated time-series was about 250 years. The proportion of deviance explained by the GLM was calculated using Dsquared() in the modEvA package[70]. We predicted the total richness and within-family richness for 2050–2300 C.E. in 50-year time steps using glm.predict() based on the most important predictor variable (alpine habitat area) in the GLM models (Supplementary Code 5). The variable importance was calculated using varImp() in caret package[71]. The within-family sedaDNA data was analysed using the same data processing.

**Reporting summary**. Further information on research design is available in the Nature Research Reporting Summary linked to this article.

## Data availability

The raw NGS sequencing data that support the findings of this study have been archived in NCBI Sequence Read Archive (SRA) with the accession code SRR13957608 and in BioProject PRJNA596631. It also has been deposited in addition to the tag-to-sample matrix and taxonomic reference database that support the NGS sequencing data analysis in Dryad Digital Repository with the identifier https://doi.org/10.5061/dryad.vdncjsxth. The filtered sedaDNA datasets analysed during this study are provided in Supplementary Data 1 and 2. The data used for statistical analyses are available in Supplementary Data 3. The AMS-dating results and calibrated ages of the lake-sediment core can be found at https://doi.org/10.1007/s00334-009-0219-5. The pollen data have been published at https://doi.org/10.1016/j.yqres.2009.12.003 and https://doi.org/10.1016/j.palaeo.2009.12.001. The gridded precipitation data sets are open access and can be downloaded from HAR (https://www.klima.tu-berlin.de/index.php?show=daten_har2&lan=en) and GPCC (https://www.dwd.de/EN/ourservices/gpcc/gpcc.html). Source data are provided with this paper.

## Code availability

The Obitools scripts for NGS sequencing analysis are archived on Dryad Digital Repository with the identifier https://doi.org/10.5061/dryad.vdncjsxth. The MATLAB code for glacier modelling is available at https://csdms.colorado.edu/wiki/Model:Gc2d. Rarefaction v1.0 is available on Zenodo: https://doi.org/10.5281/zenodo.4562708. PastElevationChange v1.0, which includes ice cover, palaeo-temperature, digital elevation models of the lake catchment, 2006–2300 C.E. temperature with RCP4.5 to reproduce the habitat simulation within lake catchment, is available on Zenodo: https://doi.org/10.5281/zenodo.4562675. The R codes for statistical analyses and visualization are available in Supplementary Code 1–5.

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

## Acknowledgements

We thank David E. Boufford for providing some information about modern terrestrial plants in Hengduan Mountains, Yaling Wu for help with sub-sampling, and Cathy Jenks for English proofreading. We also thank Xinghua Li for providing the shapefile of Hengduan Mountains. This study was funded by the Deutsche Forschungsgemeinschaft (DFG, German Research Foundation, grants 410561986 to S.K., Mi 730/1-1,2 to S.M.), US National Science Foundation (grants DEB-9705795, DEB-0321846), and China Scholarship Council (grant 201606180048 to S.L.).

## Author contributions

U.H. designed this study. U.H. and S.L. led the interpretation and writing. S.L. wrote the first draft of the manuscript. S.M. contributed to the core collection and S.L. performed lab work. S.L., K.R.S., H.H.Z., L.S.E. implemented the sedaDNA data analysis. S.L., U.H., S.K. implemented the statistical analyses. S.K. performed habitat area simulation. D.S. constructed the glacier model. R.H.R. provided modern vegetation information. All authors discussed the results and provided intellectual input to the manuscript.

## Funding

## Competing interests

All authors declare no competing interests.
