## [Peer Review File · Nature Communications]

Reviewer #1:

Remarks to the Author:

I think this is an excellent paper demonstrating once again the importance of sedDNA analysis from lakes in reconstructing flora and vegetation change through time. SedDNA show its potential in working as a local signal of flora change and a good example is shown in Figure 1 were Ericaceae and Picea is detected in accordance with the climate record but not pollen.

I think it is very interesting paper worth to be published in Nature Comm. It is well written and well presented, and with nice figures.

I do have however some few comments, see below.

Line 22

-4600 in the figure

Line 184

Why not radiocarbon ages obtained from macrofossil and only bulk organic matter samples?

Line 188-211:

How many PCR replicates did the author do and which extraction method did they use? Please add this information.

What is the reaction volume of the PCR reaction? 3 uL DNA in 20 uL or 50 uL makes a difference. I wonder also if the authors looked at inhibition before PCR. In our experience this is highly correlated with nr of reads in metabarcoding and can have importance in biodiversity estimates (abundance of species).

In our experience full glacial sediments show no inhibition at all and larger number of reads. There seems however not to be a strict correlation between inhibition and carbon content in our sediments since although inhibition increase through the Holocene it also decreases with increasing carbon content.

It is difficult to say whether the results of this study are biased by inhibition, and the only way is to test for it (which was not done), but we notice that nobody part from us is doing this at the moment, nor it is discussed. A way to check is to check if you find massive differences in total read counts between samples. If there is a big differences than there might be inhibition going on, if everything looks relatively uniform than inhibition might not be a problem. Anyway, I suggest the issue being at least discussed here.

Line 212-226: I wonder how many of the plant species on the Tibetan Plateau have been sequenced? It is a biodiversity hotspot so the authors have a database gap. So how are you sure are they that you did not group closely related species into a single mOTU? In other words, how confident are the authors that counted DNA species is a reliable indicator for biodiversity at the study site? There might also be a problem that it is very difficult to prove true absence of species. Worth discussing this point too in my opinion.

Line 272-274: Was the rarefaction done on total read count or after assignment to taxonomy? If is done after assignment, then there might be a bias towards sequences that are in the reference database?

Line 274-276: not sure I get what the authors mean here.

Line 278: Relative abundance of what? Reads or species in the repeats? If its relative abundance of

reading counts I'm not sure that even rarified are a good indicator of abundance. And as far as I know, for any Alpha and Beta diversity, there is an abundance component.

Line 280-300: I don't have enough knowledge to say much about the diversity studies done here and I cannot comment.

The use of *Picea* as a signal of land-use change I think is a good choice here since trnL g/h fragment cannot distinguish between *Picea abies*, and the two species in the study area: *Picea likiangensis*, and *Picea purpurea*, all their sequences are identical.

Reviewer #2:

Remarks to the Author:

The author(s) have generated a new sedimentary ancient DNA (sedaDNA) record from a previously-studied lake in Tibet, in order to test drivers of alpine plant diversity within the system. The novelty here is using the sedaDNA data (plant species richness) both to test drivers of change and to use these findings to estimate the impact of future climate changes on the local plant species richness. Such an approach has huge potential to be used in other regions and, potentially, across broader taxonomic groups.

The manuscript is generally well-written, flows logically, and the methods are mostly detailed enough to enable replication. However, I identify several main issues concerning data quality and the statistical parameters used. Although addressing these may change the conclusions, the novelty described above would remain intact. I therefore recommend a major revision.

Main:

The authors performed two PCR replicates per sample, which were then pooled, and all results are reported on a per-sample basis. However, how consistent were the two PCR replicates within samples? Looking at Supplementary Data 1, it seems that some PCR replicates may have completely dropped out. Were these not sequenced? (eluded to on line 207-208). As the likelihood of detecting low-template taxa increases with the number of PCR replicates, not having two positive PCR replicates would reduce the richness observed on a per sample basis. Worryingly, this seems to particularly affect batch B for ESL024-ESL027, ESL029, which are samples with lowest richness ~14 ka (Fig. 2). The authors should more thoroughly scrutinize their 'total richness' results, and test if their results are consistent were only one PCR replicate used, perhaps using the replicate that was most deeply sequenced for each sample.

line 282-285, Supplementary Table 2: it is not clear why the authors chose three overlapping periods (18-10, 14-3.6, 10-0 ka) to test their selected potential drivers of plant species richness. These periods should be non-overlapping and divided more appropriately for hypothesis testing. The four categories used for the turnover analyses (Supplementary Figure 2) would work: the impact of late-Holocene land use (line 119-120) should just use 3.6-0 ka, and the impact of early-Holocene forest expansion (line 116-118) should just use 10-3.6 ka.

line 70-71: what does "assumed land-use indicator" mean for *Rheum alexandrae*? If this was determined from the present results (ie. it is especially prevalent in the past 3.6 ka), then this taxon should not be used as a land-use indicator for any statistical analyses, as it was not known as such a priori. In addition, it would be better to show *Nardostachys jatamansi* in Fig. 2i, rather than *Rheum alexandrae*, as this taxon is known to be human-related.

line 65, 237-239: it is great to see that the authors have scrutinised their list of retained barcodes against a regional flora to remove potential false positives. However, there could still be artifacts (especially from PCR errors) in the data potentially over-inflating diversity for specific taxa. In the absence of an exhaustive local reference database, which would be required to correct for this, it would be helpful if the authors briefly expanded on why they have confidence in the retained 218 barcodes as representing 218 species. For example, were 24 species of *Pedicularis* and 8 species of *Rhododendron* realistically present in the catchment over the time period covered?

line 278, Supplementary Figure 2: The authors calculate turnover between the four main intervals in the record. However, could this be impacted by different sample sizes within each interval? I suggest the authors instead use mean (and standard deviation) turnover, immigration, and emigration between these intervals.

line 123: the authors use relative read abundance to explicitly infer relative taxonomic abundance. This relationship is very poorly understood in sedaDNA data sets, and there is good evidence to suggest that they may be unrelated. For example, the authors are using Platinum HiFi Taq (line 199), a polymerase which is known to introduce bias in relative read abundance (Nichols et al. 2018, Molecular Ecology Resources). The authors should explicitly state the limitations of using relative read abundance to infer relative taxonomic abundance.

lines 89, 258: in the methods, specify exactly which temperature proxies were used. This is eluded to in the Fig. 2 legend as 30-90 degrees N. The authors should also justify this choice, i.e. given that the lake is located at 31 degrees N, should the 30 N-30 S proxy also be tested?

line 194: the authors need to provide a list of the 8-bp tag sequences used, and which sample/PCR replicates they correspond to, otherwise the provided raw data (line 302-304) are unusable.

line 209-211: more detail is required in this section. State (1) whether a single amplicon was pool sent to FASTERIS, (2) how many libraries were prepared from this pool(s), (3) what read length was used on the HiSeq, (4) that the run was in paired-end mode, and (5) what proportion of a lane was this library(s) sequenced on.

I am assuming, based on 6 million filtered reads, that the library(s) were sequenced on part of a HiSeq-4000 lane. If that is the case, it is important to know whether there were other libraries with the same amplicon tags that were pooled on this lane. This is because the MetaFast protocol produces single indexed libraries and serious library index swapping issues have been known to impact the HiSeq-4000.

line 230-232: did these 'assumed contaminants' occur in the controls and samples, or just the samples? If the latter, these may be true sequences not represented in the reference databases that are being misidentified as contaminants. A sentence noting this should be included.

line 237: I do not agree with the author's explanation that NTC6 was affected by reagent contamination. The recovered taxa are not usual contaminants, and one would expect the samples and other controls to also be impacted were the reagents contaminated. There are other possibilities that fit the data better. First, is that the well was not completely sealed during PCR and so previously generated amplicons from the thermal cycler could enter the well during the PCR. Second, is that there was an error during the tag-to-sample demultiplexing, and that this tag actually belongs to a sample (easily checked).

The authors should add the results from all controls to the Supplementary Data files.

Minor:

line 23: as currently written, it is unclear whether "one third of the vascular plant flora of China" refers

to the entire Tibetan Plateau or the regions at ~3,600 m asl within the Plateau.
line 35: what does "habitat diversity" mean? Is it "total alpine habitat diversity"?
line 40-41: this sentence needs one or two citations.
line 52: move the lat-long and altitude data from the introduction to the methods (~line 165).
line 65: specify vascular plant
line 72-73: what does "integrated catchment signal" mean?
line 77-78: based on Supplementary Figure 2, this is not true - there is more emigration than immigration after 3.6 ka.
line 82: state also that these 984 barcodes are likely to also include PCR artifacts.
lines 96, 133: "total habitable extent" and "simulated alpine habitat extent" initially confused me, as the Fig. 2 legend states "Alpine habitat area" and "simulated total habitat area". Recommend rephrasing the legend and perhaps explicitly referring to "black line" or "red area" in the text to clarify.
line 104: what are "further processes"?
line 109: why no show the Mg/Ca ratio throughout the record in Fig. 2?
line 88-129: supplementary table 2 is referred to multiple times throughout this section. I suggest including the key results in a main text table, and keeping the full results in the supplement.
line 108-110: this result is non-significant (according to supplementary table 2), so ensure this is clearly stated.
line 128: do the authors mean "a few taxa" rather than "selected individuals"?
line 133: change "confirms" to "is consistent with", as these experimental studies were not explicitly tested.
line 135: for context and readability, state when habitat loss due to forest invasion occurred.
line 191: which DNA extraction protocol was used, and what was the mass of sediment input for extraction?
line 195-202: what was the final reaction volume?
line 207: presumably "ancient DNA separated" means "ancient DNA lab, physically separated"?
line 207: clearly state that each PCR batch was replicated once for two PCR replicates per sample.
line 208: define "NTC".
line 207-208: what does "positive sample PCR products" mean? Only those with visible bands on a gel?
line 223: add additional citations for ArctBorBryo (Willerslev et al. 2014, Nature; Soininen et al. 2015, PLoS One).
line 241: state that one sample (ESL024) was dropped, due to no taxa detected.
line 272: rephrase "sample size" as "sequencing depth" or similar.
line 282: remove "Pleistocene-Holocene transition" as this is at 11.7 ka, not 18-10 ka.
line 297-299: what about human/grazing indicators?
line 464: are the CONISS clusters based on taxonomic composition or relative read abundance?
line 486: specify this is "relative read abundance"
line 502: invert x-axis of Supplementary Figure 4 to be consistent with other plots.
line 512: this could be a single plot with different colored lines, as the current plots are misleading due to the different y-axis ranges.
Supplementary Table 1: divide the "identification ability" values by 100.
Supplementary Table 2: replace "non-sig" with the correct alpha level. Suggest highlighting rows with a significant result in bold to enhance readability.
line 530: should "sFamily" be "sRichness"? What does "Aa" mean?

Typos/grammer:

line 6: instead of two abbreviations, perhaps just say "18,000 years".
line 95 and throughout: replace "insignificant" with "non-significant".
line 137: correct spelling of "conservation".
line 167-168: change "glacier activities" to "glacial activity".
line 208: change "having a" to "including the".
line 233: correct "Table " to "Data".

lines 253, 256, etc: correct citations.
line 288: change "check significance" to "check the significance".
line 477: change "are" to "were"
line 451: delete "areal".
line 508: delete "in" and change "band across the" to "bands across".

Reviewer #3:
Remarks to the Author:

This manuscript entitled 'Alpine habitat loss threatens the future plant diversity of the Tibetan Plateau' applied a time-series approach to infer past richness from sedimentary ancient DNA of one core from a catchment area in the south eastern Tibetan Plateau over the last 18 ka (cal ka BP). Based on the established relationship between plant richness and environmental changes, especially changes of the 'alpine habitats', the authors concluded that a simulated alpine habitat loss in a warmer future could cause a 41% decrease in plant richness at the study site over the next 250 years. The MS was well written but has some unclear statements and uncertain speculations. This work well described the historical changes in plant richness and vegetation composition with a relatively high resolution. The sedimentary ancient DNA method showed an advanced and practical way to infer such historical changes across a long time span. However, the results contain a lot of speculations due to scale issue, which makes conclusions not so convincing. To date, both field observations and model predictions in either European Alps or Asia mountains (also suggested by Liang et al 2018, J BIOGEOGR, the reference 5) indicated that upward shift of alpine plant species would result in high species richness in the context of climate warming. Though the authors drew conclusions challenging these findings, I did not see any superior approaches of this study compared with traditional methods, such as SDMs, to make the story plausible. The major problem is that the authors imposed coarse scale relationship established between species richness and environmental change from history on predicting fine scale biodiversity change in the future. In other words, the spatial-temporal scale throughout the analysis was not consistent at several dimensions. Here I listed several issues that the authors did not clearly address in the MS.

Major comments

- (1) Macrorefugia vs microrefugia. I think alpine habitat extent in this study somehow refers to macrorefugia. If this is the case, the authors should refer to the paper from THEOFANIA et al 2014 GCB, from which it clearly showed the importance of supporting functions of microrefugia when forecasting the fate of alpine plant species under climate change. In this work, I did not see any concerns on this issue.
- (2) To what extent, the information derived from only one core could be extrapolated to a large scale? Alpine ecosystem is featured by high heterogeneous landscape and rugged terrain. Is it confident that one core could capture the spatial environmental variations (altitudinal and horizontal) of this alpine ecosystem? Would it benefit from more cores since Naleng lake is not the only lake in this study area as indicated from your previous work (reference 34)?
- (3) The established relationship between species richness and historical environmental change of the past 18 ka could be used to predict future change of the next 250 years? You divided the past 18 ka into several stages, the time span of each stage far exceeds 250 years. Though the authors included simulation of the future change of alpine habitat extent, it seems the constructed relationship at coarse temporal scale was imposed to predict future changes at fine temporal scale, which is very difficult to understand.
- (4) The simulation of alpine habitat extent. First, I find it very difficult to interpret how climate changed during the simulation period. Why climate data of a meteorological station 80 km away at 50 m a.s.l. was applied to calculate the temperature of a mountainous area with elevation ranging from 1000 m to 6000 m a.s.l.? Second, how treeline changed during the simulation period, which process tightly linked to the area above treeline? Thirdly, according to Supplementary Figure 5, does it mean

by the year 2300, alpine habitats above treeline was almost lost? due to forest invasion? Again, since it is not clear how treeline responds to the environmental change in this area, such result needs more evidences. Finally, it is also very uncertain how species richness followed the habitat change in the next 250 years. For example, if alpine habitat extent decreased by 10%, how species richness responded to such change? It seems there were too many speculations from this part.

(5) The number of plant species identified by sedaDNA was obviously much lower than that of current species pool in this study area. Despite of biological interaction and evolutionary adaption of alpine plant species, is the identified number of historical plant species large enough to predict future biodiversity change?

Minor comments

There are some unclear statements through the MS. Note that I did not list them all.

(1) The authors argued methodological limitations of the traditional space-for-time approaches.

However, I did not find any comparison or justification on how superior their methodology is.

(2) Line 156, what do you mean by protect? I did not see any practical suggestions to 'protect' the upper alpine habitat.

(3) Line 246, how the numerical ice flow model GC2D worked? Needs descriptions. If it is only about how glacier retreated during the past 18 ka, then I doubt its application to simulate changes of alpine habitats since plant establishment (e.g., treeline species) after glacier retreat related to seed dispersal, available microhabitats, and climate, etc.

(4) Line 445, better if location of the catchment b) to be shown in a)

(5) Line 466, the landscape change is true or just speculated, or from the GC2D? If it is not true, I would prefer a table instead.

(6) Line 479, Figure 4. No surprising extremely low species richness if there is no alpine habitat existed. Again, how changes of alpine habitat extent led to the species richness change is not clear.

(7) Line 515, commonly, sedaDNA obtained high taxonomic resolution but identified less taxa than pollen, which is contracted to Supplementary Table 1. It won't hurt if authors explained this a little bit.

The responses are in *blue*. The revisions are marked in *red* in the revised manuscript. The comments were separated into several parts and responded to point by point.

Reviewers' comments:

Reviewer #1 (Remarks to the Author):

I think this is an excellent paper demonstrating once again the importance of sedDNA analysis from lakes in reconstructing flora and vegetation change through time. SedDNA show its potential in working as a local signal of flora change and a good example is shown in Figure 1 where Ericaceae and Picea is detected in accordance with the climate record but not pollen. I think it is very interesting paper worth to be published in Nature Comm. It is well written and well presented, and with nice figures.

I do have however some few comments, see below.

Response: We are thankful to Reviewer #1 for the constructive comments.

Line 22

-4600 in the figure

Response: We are sorry for the misunderstanding. The 4,600 m a.s.l. is the peak of area distribution within Lake Naleng catchment (Fig. 1b, bar plot). In line 23, we refer to the peak of plant species within Hengduan Mountains at 3,600 m a.s.l. (Fig. 1a, red dotted line). To highlight this information, we now refer to corresponding curve.

Line 184

Why not radiocarbon ages obtained from macrofossil and only bulk organic matter samples?

Response: Our group dated the bulk organic carbon because no macrofossils were available throughout the core.

New text line 231: “As macrofossils were absent throughout the core, sixteen samples of bulk organic carbon were selected for accelerator mass spectrometer (AMS) ¹⁴C dating at the Leibniz Institute Kiel.”

Line 188-211:

How many PCR replicates did the author do and which extraction method did they use? Please add this information.

Response: There were two PCR replicates for each lake-sediment sample. We used the PowerMax[®] Soil DNA Isolation kit (Mo Bio Laboratories, Inc. USA) with a partially modified protocol.

New text lines 238-245: “Each extraction batch included nine samples (3-10 g sample⁻¹) and one extraction control, which was treated with a partially modified protocol of PowerMax[®] Soil DNA Isolation kit (Mo Bio Laboratories, Inc. USA). The isolation of DNA was first processed by loading 15 mL PowerBead solution, 1.2 mL C1 buffer, 0.8 mg proteinase K (VWR International), 0.5 mL 1 M dithiothreitol (VWR International) and samples into PowerBead tubes. Then, all tubes were vortexed in 10 min and incubated at 56°C in a rocking shaker overnight under the aluminium foil protection. The subsequent extraction steps followed the manufacturer’s instructions of the kit and were completed on the second day.

New text lines 262-264: “Each PCR batch was replicated until obtaining two positive PCR replicates for each lake-sediment sample when the associated controls were negative.”

What is the reaction volume of the PCR reaction? 3 uL DNA in 20 uL or 50 uL makes a difference.

Response: The PCRs were set up in a total of 25 µL per reaction (please see line 250).

I wonder also if the authors looked at inhibition before PCR. In our experience this is highly correlated with nr of reads in metabarcoding and can have importance in biodiversity estimates (abundance of species).

In our experience full glacial sediments show no inhibition at all and larger number of reads. There seems however not to be a strict correlation between inhibition and carbon content in our sediments since although inhibition increase through the Holocene it also decreases with increasing carbon content.

It is difficult to say whether the results of this study are biased by inhibition, and the only way is to test for it (which was not done), but we notice that nobody part from us is doing this at the moment, nor it is discussed. A way to check is to check if you find massive differences in total read counts between samples. If there is a big differences than there might be inhibition going on, if everything looks relatively uniform than inhibition might not be a problem. Anyway, I suggest the issue being at least discussed here.

Response: During the genetic laboratory work, we did not check the inhibition before the PCR amplification. Unfortunately, we cannot perform these analyses at the moment. However, following your suggestion, we investigated the potential impact of inhibition on richness by correlating read count with richness. Our results indicate that (1) excluding 5 outliers we have rather similar read counts (Supplementary Figure 1b); (2) during the late glacial we have as well low and high read counts; (3) we found a very weak positive non-significant correlation between read count and richness; and (4) total organic carbon (TOC) of Lake Naleng showed no relationship with either richness or read counts (Figure #1.1). We thus conclude that inhibition during PCR did not impact the richness signal. (We note that this discussion was only partly transferred to the manuscript because we did not find a study that discussed the potential impact of inhibition on richness related to TOC.)

Figure #1.1 The total sequence count for each sample (grey bars), TOC (blue line, Opitz et al., 2015) and total plant taxa richness (orange line).

New text lines 70-73: “Apart from 5 outliers with high read numbers, PCR replicates yielded read counts of a similar order of magnitude (Supplementary Figure 1b). We found no correlation between read counts and total plant taxa richness ($df = 69$, $\rho = 0.014$, $p = 0.908$). From this we conclude that read counts has no impact on the inferred diversity signal.”

New figure lines 648-655: “Supplementary Figure 1 | Information about *seadDNA* data with 100% best identity and plant taxa richness for sediments of Lake Naleng.”

Opitz et al. Climate variability on the south-eastern Tibetan Plateau since the Lateglacial based on a multiproxy approach from Lake Naleng – comparing pollen and non-pollen signals. Quaternary Science Reviews 115, 112–122 (2015).

Line 212-226: I wonder how many of the plant species on the Tibetan Plateau have been sequenced? It is a biodiversity hotspot so the authors have a database gap. So how are you sure are they that you did not group closely related species into a single mOTU? In other words, how confident are the authors that counted DNA species is a reliable indicator for biodiversity at the study site? There might also be a problem that it is very difficult to prove true absence of species. Worth discussing this point too in my opinion.

Response: We agree with the reviewer that we are dealing with a geographic area in which we indeed face a database gap, and we unfortunately cannot rely on a specific regional database of sequences for our marker. Therefore, we cannot be certain that our taxonomic assignments to taxa and genus levels are correct, and we might well be underestimating the species richness per sample. However, we have further evaluated the reliability of our richness pattern through time. We found a similar pattern of total plant taxa richness is obtained regardless of which dataset we use, i.e. we compared the richness obtained with the dataset before taxonomic assignment, the dataset containing all terrestrial seed plant sequences, the dataset with best identity 0.95 and the dataset with best identity 1. It is thus reasonable to assume non-completeness of the reference database did not affect the temporal sedaDNA richness pattern.

New text lines 100-104: “Furthermore, the similar temporal pattern of total plant taxa richness was obtained when analysing the data before taxonomic assignment and the data containing all terrestrial seed plant sequences (Methods), as indicated by the highly significant correlations of these time-series with total plant taxa richness (Supplementary Table 1).”

New Supplementary Table: lines 688-700

Supplementary Table 1 | The correlation of plant taxa richness based on different datasets indicates that the temporal variations of total plant taxa richness were accurately reflected by the data with best identity 1 in this study

	rho	p-value	df
Bestid 1 vs. Non-ecotag data	0.768	5.38E-15	69
Bestid 1 vs. terSeq data	0.830	3.52E-19	69
Bestid 1 vs. Bestid 0.95	0.888	6.46E-25	69
Bestid 1 vs. Single data	0.951	7.74E-37	69

Bestid 1: after further sequence filtering, collected those terrestrial seed plant sequences that having best identity value of 1 and present in two PCRs at least.

Bestid 0.95: after further sequence filtering, collected those terrestrial seed plant sequences that having best identity value ≥ 0.95 and present in two PCRs at least.

terSeq: the data containing all terrestrial seed plant sequences without further sequence filtering.

Non-ecotag: the data before taxonomic assignment

Single data: based on data Bestid 1, only using deeply sequenced PCR replicate for each lake-sediment sample.

rho: Spearman's Rank correlation coefficient
df: degrees of freedom
For detail information, please see Methods.

Line 272-274: Was the rarefaction done on total read count or after assignment to taxonomy?
If is done after assignment, then there might be a bias towards sequences that are in the reference database?

Response: In our study, the rarefaction was done after taxonomic assignment. Following your suggestion, we also rarefied the dataset before taxonomic assignment to its minimal read counts (16,209) and calculated taxa richness. We found a similar temporal pattern of total plant taxa richness. Accordingly, we conclude that the variations of taxa richness are not significantly affected by rarefying before or after taxonomic assignment.

New text lines 100-104: “Furthermore, the similar temporal pattern of total plant taxa richness was obtained when analysing the data before taxonomic assignment and the data containing all terrestrial seed plant sequences (Methods), as indicated by the highly significant correlations (Supplementary Table 1).”

New Supplementary Table: lines 688-700

Supplementary Table 1 | The correlation of plant taxa richness based on different datasets indicates that the temporal variations of total plant taxa richness were accurately reflected by the data with best identity 1 in this study

	rho	p-value	df
Bestid 1 vs. Non-ecotag data	0.768	5.38E-15	69
Bestid 1 vs. terSeq data	0.830	3.52E-19	69
Bestid 1 vs. Bestid 0.95	0.888	6.46E-25	69
Bestid 1 vs. Single data	0.951	7.74E-37	69

Bestid 1: after further sequence filtering, collected those terrestrial seed plant sequences that having best identity value of 1 and present in two PCRs at least.

Bestid 0.95: after further sequence filtering, collected those terrestrial seed plant sequences that having best identity value ≥ 0.95 and present in two PCRs at least.

terSeq: the data containing all terrestrial seed plant sequences without further sequence filtering.

Non-ecotag: the data before taxonomic assignment

Single data: based on data Bestid 1, only using deeply sequenced PCR replicate for each lake-sediment sample.

rho: Spearman's Rank correlation coefficient

df: degrees of freedom

For detail information, please see Methods.

New text lines 358-364: “To investigate potential methodological biases of the *sedaDNA*-based plant taxa richness, two additional datasets were set up: (1) metabarcoding data before taxonomic assignment (hereafter ‘non-ecotag’); (2) containing all terrestrial seed plants

sequences without further sequence filtering (hereafter ‘terSeq data’). Both datasets were rarefied to their respective minimal total read count (16,209; 14,645) 100 times. We investigated whether plant taxa richness is correlated to read counts and whether it is correlated with total plant taxa richness of “bestid1” dataset using “*corr.test(adjust = “none”)*”.”

Line 274-276: not sure I get what the authors mean here.

Response: We have revised these sentences as below:

New text lines 375-379: “In order to calculate the plant taxa richness for each single taxonomic family, we divided *seDaDNA* sequence data into subsets of taxonomic families and rarefied these subsets to a cut-off value of 100 total read counts to minimize the effect of relative abundance of taxonomic family. Such richness signals of families with low read counts (e.g. Ranunculaceae) could be compared with richness signals of families having higher read counts (e.g. Asteraceae).”

Line 278: Relative abundance of what? Reads or species in the repeats? If its relative abundance of reading counts. I'm not sure that even rarefied are a good indicator of abundance. And as far as I know, for any Alpha and Beta diversity, there is an abundance component.

Response: Thank you for your comment. We re-calculated the taxa turnover (please see the new Supplementary Figure 3). The methods have been corrected and now includes rarefaction. We agree that using abundance of metabarcoding data has some problems. Therefore, we based our main investigation on richness analyses using a presence/absence-based index. On the other hand, we found that compositional change based on abundance data generally tracks the known changes from pollen analyses as indicated in the PCA.

New text lines 381-385: We summed up the samples in each zone and computed the mean value of read counts per zone. Then, we rarefied the zonal data to its minimal total read count (40,377) 100 times. Afterwards, we computed the total vegetation turnover (beta diversity) using the “*turnover()*” function in the “*codyn*” package⁶⁸ based on the rarefied zonal data (Supplementary Code 2).”

68. Hallett, L. M. et al. *codyn: An R package of community dynamics metrics. Methods Ecol. Evol.* 7, 1146–1151 (2016).

Line 280-300: I don't have enough knowledge to say much about the diversity studies done here and I cannot comment.

Response: We appreciate your other constructive suggestions.

The use of *Picea* as a signal of land-use change I think is a good choice here since trnl g/h fragment cannot distinguish between *Picea abies*, and the two species in the study area: *Picea likiangensis*, and *Picea purpurea*, all their sequences are identical.

Response: Thank you for your comment. We think *Picea* would be a good indicator of land-use change at lower elevations where humans clear areas for cultivation. Lake Naleng is in a high-elevation region (4,200 m a.s.l.). Past human activity in the Lake Naleng catchment was likely restricted to nomadic pastoralism as is the case today, which does not of course exclude that they made use of wood. However, we have no evidence for intense land use and the closest known archaeological site (Kharub, a Neolithic site at 3,100 m a.s.l.) is 230 km west of our study site. Furthermore, a previous study (exploring the charcoal signals from Lake Naleng record) found no evidence for increasing human-induced forest fires in the late Holocene (Kramer et al., 2010). Thus, we assume that the forest shifts can be attributed to late-Holocene climate cooling rather than to human activities in our study area.

New text lines 147-149: “We assume that the retreat of forests is related to late Holocene cooling and weakening of the Asian summer monsoon, not to human impact, and is supported by a lack of late Holocene forest burning²⁰.”

New text lines 163-164: “We assume that the signs for late Holocene grazing intensification in *sedaDNA* and pollen records (Supplementary Figure 2) are related to human impact.”.

20. Kramer, A., Herzschuh, U., Mischke, S. & Zhang, C. Holocene treeline shifts and monsoon variability in the Hengduan Mountains (southeastern Tibetan Plateau), implications from palynological investigations. *Palaeogeogr. Palaeoclimatol. Palaeoecol.* **286**, 23–41 (2010).

The responses are in blue. The revisions are marked in red in the revised manuscript. The comments were separated into several parts and responded to point by point.

Reviewer #2 (Remarks to the Author):

The author(s) have generated a new sedimentary ancient DNA (sedaDNA) record from a previously-studied lake in Tibet, in order to test drivers of alpine plant diversity within the system. The novelty here is using the sedaDNA data (plant species richness) both to test drivers of change and to use these findings to estimate the impact of future climate changes on the local plant species richness. Such an approach has huge potential to be used in other regions and, potentially, across broader taxonomic groups.

The manuscript is generally well-written, flows logically, and the methods are mostly detailed enough to enable replication. However, I identify several main issues concerning data quality and the statistical parameters used. Although addressing these may change the conclusions, the novelty described above would remain intact. I therefore recommend a major revision.

Response: We thank reviewer #2 for the constructive comments.

Main:

The authors performed two PCR replicates per sample, which were then pooled, and all results are reported on a per-sample basis. However, how consistent were the two PCR replicates within samples? Looking at Supplementary Data 1, it seems that some PCR replicates may have completely dropped out. Were these not sequenced? (eluded to on line 207-208). As the likelihood of detecting low-template taxa increases with the number of PCR replicates, not having two positive PCR replicates would reduce the richness observed on a per sample basis. Worryingly, this seems to particularly affect batch B for ESL024-ESL027, ESL029, which are samples with lowest richness ~14 ka (Fig. 2).

Response: Read numbers in most PCR replicates are consistent, as Supplementary Figure 1a indicates (also shown in Figure #2.1). These PCRs were sequenced but yielded no results.

First, we did not include sample ESL024 in statistical analyses because no sequence was detected in ESL024a and b (dated to 14.3 ka). We have indicated this in lines 305-306. We assume that glacial melting was likely very strong during the Bölling/Alleröd period and

the high sedimentation rate may have diluted the plant matter concentration in the sediments. However, we did not find any ecological evidence or mistakes in the lab work to explain the low DNA in these replicates. Accordingly, it was excluded from the statistical analyses.

Figure #2.1 The total read count for each PCR replicate of lake sediment. [also shown in Supplementary Figure 1a, lines 648-655]

Second, we also considered whether samples having a single replicate (ESL025-ESL027, ESL029 and ESL065) may have affected plant taxa richness, i.e. low richness in samples with low read counts. However, we found only a very weak non-significant positive correlation between read counts and total plant taxa richness ($df = 69$, $\rho = 0.014$, p -value = 0.908). Therefore, we conclude that taxa richness is not markedly influenced by the read counts.

New text lines 305-306: “Sample ESL024 was excluded from further statistical analysis because no reads were obtained (Supplementary Data 1 and Data 2).”

New text lines 70-73: “Apart from 5 outliers with high read numbers, PCR replicates yielded read counts of a similar order of magnitude (Supplementary Figure 1b). We found no correlation between read counts and total plant taxa richness ($df = 69$, $\rho = 0.014$, $p = 0.908$). From this we conclude that read count has no impact on the inferred diversity signal.”

The authors should more thoroughly scrutinize their 'total richness' results, and test if their results are consistent were only one PCR replicate used, perhaps using the replicate that was most deeply sequenced for each sample.

Response: Thank you for your advice. Following your suggestion, we first selected the replicate with the highest read count for each sample (hereafter referred to as “single data”, 71

replicates), and then rarefied the single data to its minimal read count (base count = 6,339) 100 times to calculate the taxa richness (hereafter referred to as “single taxa richness”). Afterwards, we compared the single taxa richness with the total taxa richness that was calculated including all replicates of a sample (138 replicates, as shown in the manuscript). They show a very similar pattern, as indicated in Figure #2.2, supported by a high positive correlation ($df = 69$, $\rho = 0.951$, $p = 7.74E-37$, Supplementary Table 1). Furthermore, we found that the compositional sequence signals do not differ between results from single PCR samples or from pooled-PCR samples, marked by the extremely good fit between samples and taxa scores for the first two PCA axes for the total data and single data (samples: $\rho = 0.986$, $p = 0.001$; taxa: $\rho = 0.993$, $p = 0.001$, Supplementary Table 2). We thus conclude that the number of replicates included in a sample do not affect the richness signals and our compositional signal used all available replicates in our analyses.

Figure #2.2 Comparison of the replicates-based and single replicate-based plant taxa richness over the past ~18,000 years [also shown in lines 648-655: Supplementary Figure 1 | Information about sedaDNA data with 100% best identity and plant taxa richness for sediments of Lake Naleng. c, Plant taxa richness was calculated based on “bestid1” data containing all positive PCR replicates. d, Plant taxa richness was computed based on single data, consisting of deeply sequenced PCR products that have a higher total read count within each sample. Before calculation, the sample ESL024 was excluded from both datasets due to no plant *sedaDNA* in both its PCR replicates.]

New text lines 74-78: “Also, neither variations in plant taxa richness nor compositional signals differed between results from single PCR samples or from pooled-PCR samples (Supplementary Figure 1 c, d, Supplementary Table 1 and Supplementary Table 2).

Accordingly, we assume that the variations of plant taxa richness over time can be reliably tracked by pooling results from PCR replicates of one horizon.”

New text lines 365-374: “To test if the plant taxa richness and composition are stable, we first collected the deeply sequenced PCR product for each lake sediment sample from the dataset “bestid1” (hereafter referred as “single data”). Then, we rarefied these data 100 times based on the minimal read count (6,339) across all samples and calculated the plant taxa richness. The correlation between plant taxa richness from data “bestid1” and “single data” was calculated using “*corr.test(adjust = “none”)*”. Finally, Procrustes and Protest analyses were applied to check whether samples scores and taxa scores of the first two PCA (principal component analysis) axes of the “single data” match those of “bestid1” dataset. Only those sequences with a minimum of 0.25% were kept. Double-square root transformation was applied before PCA analysis⁶⁶. The “*procrustes()*”, “*protest()*” and “*rda(scale = FALSE)*” are available in the “vegan” package⁶⁷.

66. Zimmermann, H. H. et al. Sedimentary ancient DNA and pollen reveal the composition of plant organic matter in Late Quaternary permafrost sediments of the Buor Khaya Peninsula (north-eastern Siberia). *Biogeosciences* **14**, 575–596 (2017).

67. Oksanen, J. et al. *vegan: Community Ecology Package*. (2019).

New Supplementary Table: lines 688-700

Supplementary Table 1 | The correlation of plant taxa richness based on different datasets indicates that the temporal variations of total plant taxa richness were accurately reflected by the data with best identity 1 in this study

	rho	p-value	df
Bestid 1 vs. Non-ecotag data	0.768	5.38E-15	69
Bestid 1 vs. terSeq data	0.830	3.52E-19	69
Bestid 1 vs. Bestid 0.95	0.888	6.46E-25	69
Bestid 1 vs. Single data	0.951	7.74E-37	69

Bestid 1: after further sequence filtering, collected those terrestrial seed plant sequences that having best identity value of 1 and present in two PCRs at least.

Bestid 0.95: after further sequence filtering, collected those terrestrial seed plant sequences that having best identity value ≥ 0.95 and present in two PCRs at least.

terSeq: the data containing all terrestrial seed plant sequences without further sequence filtering.

Non-ecotag: the data before taxonomic assignment

Single data: based on data Bestid 1, only using deeply sequenced PCR replicate for each lake-sediment sample.

rho: Spearman's Rank correlation coefficient

df: degrees of freedom

For detail information, please see Methods.

New Supplementary Table 2: lines 701-707

Supplementary Table 2 | Overview of Procrustes and Protest analyses

	p-value	r	m ¹²	rmse

Bestid1 vs. Single data: samples	0.001	0.986	0.027	0.020
Bestid1 vs. Single data: taxa	0.001	0.993	0.014	0.014

Bestid 1: after further sequence filtering, collected those terrestrial seed plant sequences that having best identity value 1 and present in two PCRs at least.

Single data: based on data Bestid 1, only using deeply sequenced PCR replicate for each lake-sediment sample.

r: Correlation in a symmetric Procrustes rotation

m¹²: Procrustes sum of squares

rmse: Procrustes root mean squared error

For detail information, please see Methods.

line 282-285, Supplementary Table 2: it is not clear why the authors chose three overlapping periods (18-10, 14-3.6, 10-0 ka) to test their selected potential drivers of plant species richness. These periods should be non-overlapping and divided more appropriately for hypothesis testing. The four categories used for the turnover analyses (Supplementary Figure 2) would work: the impact of late-Holocene land use (line 119-120) should just use 3.6-0 ka, and the impact of early-Holocene forest expansion (line 116-118) should just use 10-3.6 ka.

Response: Thank you for your comment. We considered your suggestion to search for drivers within periods of similar vegetation composition (i.e. the different zones obtained from clustering) but we finally did not follow your suggestion. Our arguments for keeping our approach are as follows. Using CONISS we identified vegetation zones, i.e. major changes in the vegetation record, occurring in the transition between these zones. These compositional changes were largely tracked by richness changes. Hence, an analysis of drivers should cover the transition between the zones and not be performed within the zones. We furthermore considered applying correlation analyses between driver variables and the full time-series. However, because the relevance of drivers likely changed through time, we decided to perform correlation analyses using time-series of consecutive zones. We provide more explanation for the rationale of our approach in the text.

New text lines 386-392: “To identify the main drivers of plant taxa richness, we calculated the correlation coefficient between plant taxa richness and driver variables using “*corr.test(adjust = “Bonferroni”)*”. We separated the complete richness time-series into three time-intervals each consisting of two consecutive vegetation zones (according to CONISS), i.e. 18–10 ka, 14–3.6 ka, 10–0 ka. This approach accounts for variation of driver importance throughout the record and can even reveal sign changes in the relationship between driver variables and plant taxa richness.”

line 70-71: what does "assumed land-use indicator" mean for *Rheum alexandrae*? If this was determined from the present results (ie. it is especially prevalent in the past 3.6 ka), then this

taxon should not be used as a land-use indicator for any statistical analyses, as it was not known as such a priori. In addition, it would be better to show *Nardostachys jatamansi* in Fig. 2i, rather than *Rheum alexandrae*, as this taxon is known to be human-related.

Response: Following your suggestion, we summed up the relative read abundance of *Nardostachys jatamansi* and *Sanguisorba officinalis* as known land-use indicators. The results are very similar to those before; hence, no adjustments were necessary with respect to inferences.

New text lines 83-84: “*Sanguisorba officinalis* (Fig. 2i, grey area)²⁰ and *Nardostachys jatamansi* (a traditional Tibetan medicinal plant²³) (Fig. 2i, black line).”

New text lines 394-395: “We used ‘*gam()*’ for land-use data smoothing as it included a large number of zeros.”

20. Kramer, A., Herzschuh, U., Mischke, S. & Zhang, C. Holocene treeline shifts and monsoon variability in the Hengduan Mountains (southeastern Tibetan Plateau), implications from palynological investigations. *Palaeogeogr. Palaeoclimatol. Palaeoecol.* **286**, 23–41 (2010).

23. Singh, U.M., Gupta, V., Rao, V.P., Sengar, R.S. & Yadav, M.K. A review on biological activities and conservation of endangered medicinal herb *Nardostachys jatamansi*. *Int. J. Med. Aromat. Plant*, **3**(1), pp.113-124 (2013).

line 65, 237-239: it is great to see that the authors have scrutinised their list of retained barcodes against a regional flora to remove potential false positives. However, there could still be artifacts (especially from PCR errors) in the data potentially over-inflating diversity for specific taxa. In the absence of an exhaustive local reference database, which would be required to correct for this, it would be helpful if the authors briefly expanded on why they have confidence in the retained 218 barcodes as representing 218 species. For example, were 24 species of *Pedicularis* and 8 species of *Rhododendron* realistically present in the catchment over the time period covered?

Response: Thank you for your comment. We have indicated that the plant taxa richness can be regarded as a semi-quantitative taxa richness assessment in our study due to the lack of a local taxonomic reference database and potential errors (e.g. PCR, sequencing errors and chimeras). The Hengduan Mountains is a biodiversity hotspot. The Orobanchaceae and Ericaceae are two big common families in the Hengduan Mountains (Yu et al. 2020). For example, approximately 300 taxa of *Pedicularis* (Orobanchaceae) are endemic to the Hengduan Mountains (Meek et al.

2020). Moreover, *Rhododendron* is a large genus in the Himalaya–Hengduan Mountains (about 317 spp., Yan et al., 2015). As such, with our metabarcoding approach we face the problem of underestimation of taxa rather than of overestimation. Accordingly, we focus our analyses on patterns rather than on absolute richness inferences. Furthermore, the pattern of plant taxa richness is quite similar when comparing the richness time-series using different best identity thresholds (e.g. best identity = 100% vs. = 95%, $df = 69$, $p = 6.46E-25$, $\rho = 0.888$), suggesting that the taxa richness signal is not sensitive to the database. Accordingly, it is reasonable to assume that the sedaDNA metabarcoding approach captured the effective signal of plant taxa richness over time but likely underestimates absolute richness due to a lack of marker specificity and a specific regional taxonomic reference database.

New text lines 100-111: “Furthermore, the similar temporal pattern of total plant taxa richness was obtained when analysing the data before taxonomic assignment and the data containing all terrestrial seed plant sequences (Methods), as indicated by the highly significant correlations of these time-series with total plant taxa richness (Supplementary Table 1). Sample processing-related errors (e.g. PCR and sequencing) may have slightly inflated richness. However, we assume that we rather underestimate richness because of non-specificity of the marker and non-completeness of the reference database, which likely means that the *sedaDNA* detected taxa number is lower than the absolute taxa number recorded in the flora. Additionally, taxa rich-families²⁵ in the flora including Asteraceae, Saxifragaceae, and Orobanchaceae have highest richness in our record. Thus, plant taxa richness (total plant taxa richness and taxa richness within dominant alpine families) can be regarded as a semi-quantitative proxy of taxa richness in our study.”

Meek et al. 2020. *Phylogeography and conservation of Pedicularis (Orobanchaceae) in the Hengduan Mountains of SW China and Tibet*. doi: <https://doi.org/10.7916/d8-5yjr-kk29>. (Abstract is available. Full paper will be available starting 2022-06-05.)

Yan et al. 2015. *DNA barcoding of Rhododendron (Ericaceae), the largest Chinese plant genus in biodiversity hotspots of the Himalaya–Hengduan Mountains*. *Molecular ecology resources*, 15(4), pp.932-944.

25. Yu, H. et al. *Contrasting Floristic Diversity of the Hengduan Mountains, the Himalayas and the Qinghai-Tibet Plateau Sensu Stricto in China*. *Front. Ecol. Evol.* 8, (2020).

line 278, Supplementary Figure 2: The authors calculate turnover between the four main intervals in the record. However, could this be impacted by different sample sizes within each

interval? I suggest the authors instead use mean (and standard deviation) turnover, immigration, and emigration between these intervals.

Response: Thank you for your suggestion that we have now implemented. We re-calculated the taxa turnover (please see the new Supplementary Figure 3).

New Supplementary Figure: lines 663-665

Supplementary Figure 3 | Proportional plant taxa turnover since 18 ka. Taxa gain is defined as the proportion of immigrants that appear in the lake catchment between the selected time periods, while taxa lost is the proportional disappearance of species.

New text lines 381-385: “We summed up the samples in each zone and computed the mean value of read counts per zone. Then, we rarefied the zonal data to its minimal total read count (40,377) 100 times. Afterwards, we computed the total vegetation turnover (beta diversity) using the “*turnover()*” function in the “*codyn*” package⁶⁸ based on the rarefied zonal data (Supplementary Code 2).”

68. Hallett, L. M. et al. *codyn: An r package of community dynamics metrics. Methods in Ecology and Evolution* 7, 1146–1151 (2016).

line 123: the authors use relative read abundance to explicitly infer relative taxonomic abundance. This relationship is very poorly understood in sedaDNA data sets, and there is good evidence to suggest that they may be unrelated. For example, the authors are using Platinum HiFi Taq (line 199), a polymerase which is known to introduce bias in relative read abundance (Nichols et al. 2018, Molecular Ecology Resources). The authors should explicitly state the limitations of using relative read abundance to infer relative taxonomic abundance.

Response: Thank you for your comment. We generally agree with the comment and, accordingly, do not base any major conclusion in our manuscript on results that were solely supported by abundance changes of *sed*aDNA sequence types. We now explicitly point to the problem as below:

New text lines 153-159: “Of course, the relationship between relative read abundance and relative abundance of the taxon in the vegetation is still poorly understood and previous studies indicate that biases originate from, for example, PCR setup (e.g. preference for short reads and reads with high GC content³⁰). However, studies of modern lake sediments have also shown that the compositional differences among sites are preserved^{31,32}. Similarity in compositional changes between Lake Naleng *sed*aDNA and the pollen record supports this finding (Supplementary Figure 2).”

30. Nichols, R. V. et al. Minimizing polymerase biases in metabarcoding. *Mol. Ecol. Resour.* **18**, 927–939 (2018).

31. Alsos, I. G. et al. Plant DNA metabarcoding of lake sediments: How does it represent the contemporary vegetation. *PLoS ONE* **13**, e0195403 (2018).

32. Niemeyer, B., Epp, L. S., Stoof-Leichsenring, K. R., Pestryakova, L. A. & Herzschuh, U. A comparison of sedimentary DNA and pollen from lake sediments in recording vegetation composition at the Siberian treeline. *Mol. Ecol. Resour.* **17**, e46–e62 (2017).

lines 89, 258: in the methods, specify exactly which temperature proxies were used. This is eluded to in the Fig. 2 legend as 30-90 degrees N. The authors should also justify this choice, ie. given that the lake is located at 31 degrees N, should the 30 N-30 S proxy also be tested?

Response: We used the multi-proxy temperature reconstruction provided by Marcott et al. (2013) and Shakun et al. (2012). The temperature is reconstructed based on multiple proxies (e.g. chironomid transfer functions, ice-core dD, pollen) which is now indicated in the caption of Fig. 2 (lines 591-592): “The Northern Hemisphere (30°–90°N) temperature anomaly record since last deglaciation based on multiple proxies^{56,57}”.

New text lines 307-312: “Several previous studies have shown that the climate at a millennial time-scale on the eastern Tibetan Plateau is strongly impacted by monsoons, particularly the East Asian summer monsoon, which tracks changes in the westerlies and continental warming that are largely a function of mid- to high-latitude changes^{54,55}. Thus, the past climate change in our study was inferred by the synthesized record of Northern Hemisphere (30°–90°N) temperature anomaly since the last deglaciation^{56,57}.”

54. Zhao, Y. et al. *Evolution of vegetation and climate variability on the Tibetan Plateau over the past 1.74 million years. Sci. Adv.* 6, eaay6193 (2020).
55. Herzsuh, U. et al. *Position and orientation of the westerly jet determined Holocene rainfall patterns in China. Nat. Commun.* 10, 2376 (2019).
56. Marcott, S. A., Shakun, J. D., Clark, P. U. & Mix, A. C. *A Reconstruction of Regional and Global Temperature for the Past 11,300 Years. Science* 339, 1198–1201 (2013).
57. Shakun, J. D. et al. *Global warming preceded by increasing carbon dioxide concentrations during the last deglaciation. Nature* 484, 49–54 (2012).

line 194: the authors need to provide a list of the 8-bp tag sequences used, and which sample/PCR replicates they correspond to, otherwise the provided raw data (lines 302-304) are unusable.

Response: Done.

New text lines 413-415: “The raw sequencing data, tag-to-sample matrix and ObiTools scripts for metabarcoding data analysis in this study will be uploaded to Dryad; and the DOIs will be provided after acceptance.”

line 209-211: more detail is required in this section. State (1) whether a single amplicon was pool sent to FASTERIS, (2) how many libraries were prepared from this pool(s), (3) what read length was used on the HiSeq, (4) that the run was in paired-end mode, and (5) what proportion of a lane was this library(s) sequenced on.

I am assuming, based on 6 million filtered reads, that the library(s) were sequenced on part of a HiSeq-4000 lane. If that is the case, it is important to know whether there were other libraries with the same amplicon tags that were pooled on this lane. This is because the MetaFast protocol produces single indexed libraries and serious library index swapping issues have been known to impact the HiSeq-4000.

Response: First, we have corrected the sequence machine to HiSeq 2500 after asking the sequencing company. We have added the required information as below:

New text lines 270-276: “All purified PCR products were equimolarly pooled and sent for sequencing to *Fasteris* SA, which used the MetaFast library protocol prior to sequencing on an Illumina HiSeq 2500 sequencing platform with paired-end reads of 125 bp length with the mode HiSeq High Output Version 4 by applying the HiSeq SBS Kit v4. Our project was sequenced together with another unknown sequencing project on a full HiSeq 2500 lane and resulted in 9.5 Gb with 37,922,797 generated clusters \geq Q30.”

line 230-232: did these 'assumed contaminants' occur in the controls and samples, or just the samples? If the latter, these may be true sequences not represented in the reference databases that are being misidentified as contaminants. A sentence noting this should be included.

Response: They were detected in samples not in controls except for NTC6. We have stated that the pattern of total plant taxa richness was not changed even if including them in the taxa richness analyses.

New text lines 100-104: “Furthermore, the similar temporal pattern of total plant taxa richness was obtained when analysing the data before taxonomic assignment and the data containing all terrestrial seed plant sequences (Methods), as indicated by the highly significant correlations of these time-series with total plant taxa richness (Supplementary Table 1).”

New text lines 358-364: “To investigate potential methodological biases of the *sedaDNA*-based plant taxa richness, two additional datasets were set up: (1) metabarcoding data before taxonomic assignment (hereafter referred as “non-ecotag”); (2) all terrestrial seed plant sequences without further sequence filtering (hereafter referred as “terSeq data”). Both datasets were rarefied to their respective minimal total read count (16,209; 14,645) 100 times. We investigated whether plant taxa richness is correlated to read counts for both datasets and whether it is correlated with total plant taxa richness of “bestid1” dataset using “*corr.test(adjust = “none”)*”.”

New Supplementary Table: lines 688-700

Supplementary Table 1 | The correlation of plant taxa richness based on different datasets indicates that the temporal variations of total plant taxa richness were accurately reflected by the data with best identity 1 in this study

	rho	p-value	df
Bestid 1 vs. Non-ecotag data	0.768	5.38E-15	69
Bestid 1 vs. terSeq data	0.830	3.52E-19	69
Bestid 1 vs. Bestid 0.95	0.888	6.46E-25	69
Bestid 1 vs. Single data	0.951	7.74E-37	69

Bestid 1: after further sequence filtering, collected those terrestrial seed plant sequences that having best identity value of 1 and present in two PCRs at least.

Bestid 0.95: after further sequence filtering, collected those terrestrial seed plant sequences that having best identity value ≥ 0.95 and present in two PCRs at least.

terSeq: the data containing all terrestrial seed plant sequences without further sequence filtering.

Non-ecotag: the data before taxonomic assignment

Single data: based on data Bestid 1, only using deeply sequenced PCR replicate for each lake-sediment sample.

rho: Spearman's Rank correlation coefficient

df: degrees of freedom

For detail information, please see Methods.

line 237: I do not agree with the author's explanation that NTC6 was affected by reagent contamination. The recovered taxa are not usual contaminants, and one would expect the samples and other controls to also be impacted were the reagents contaminated. There are other possibilities that fit the data better. First, is that the well was not completely sealed during PCR and so previously generated amplicons from the thermal cycler could enter the well during the PCR. Second, is that there was an error during the tag-to-sample demultiplexing, and that this tag actually belongs to a sample (easily checked).

Response: We have carefully considered this issue and agree that the sequences in NTC6 cannot be explained by a reagent contamination. First, we checked the tag-to-sample demultiplexing. It was correct according to our documentation in our lab books and in the bioinformatic demultiplexing step. Further, there was no positive gene band in the agarose gel-electrophoresis picture (please see Figure #2.3), suggesting that the cross-contamination was not likely to occur during lab work. This is supported by the fact that we preform PCR in stripe-PCR tubes with single caps, which largely reduces contamination during amplification. However, we showed by ordination analyses (please see Figure #2.4) that the NTC6 has a comparable sequence composition like a sample, that is why we largely assume that we used a different tag combination when pipetting the PCR mix for NTC6 but did not notice. This tag combination used for the NTC6 is unknown to us and we cannot retrieve sequence information for NTC6 via demultiplexing.

Figure #2.3 | The agarose gel electrophoresis picture of batch 6 (id = NTC_SL006P). Agarose gels were made 2%.

Figure #2.4 | The first two axes of a principal component analysis (PCA) of the lake sediments and NTC6 [also shown as Supplementary Figure 9 at lines 685-687]. PCR batches 4, 6 and 7 were amplified with same thermal cycler simultaneously.

Therefore, we finally decided that we do not remove the sequences detected in NTC6 from the controlled samples in the PCR batch, because we confident that it does not represent the actual NTC. We added some supplementary discussion:

New text lines 631-647: “After further sequence filtering (Methods), 32 unique sequence types (53,354 reads) were assigned to terrestrial seed plants in NTC6 in the batch with samples ESL011b-015b and ESL017b-019b. There was no positive gene band in the Agarose gel-electrophoresis, suggesting that the cross-contamination was unlikely to have occurred during the lab work. To check our supposition, we ran a principal component analysis (PCA) of all samples and NTC6. The PCA plot (Supplementary Figure 9) shows a wide difference between the sequence composition of NTC6 and the associated samples (batch 6, red dots), suggesting the origin of the contaminates in NTC 6 is unlikely to have come from this batch. Furthermore, there is no convincing evidence of cross-contamination among samples of batch 6 with batches 4 and 7 which were amplified with the same thermal cycler simultaneously, as seen by their extremely different compositional taxa. In contrast, these samples were well repeated, as suggested by the similar taxa composition of replicates (e.g. ESL011a-015a and ESL017a-019a vs. ESL011b-015b and ESL017b-019b, ESL021a-23a vs. ESL021b-23b and ESL028a vs.

ESL028b). Taken together, we think that we used a wrong tag combination while pipetting the PCR mix for NTC6 but did not notice. This tag combination is unknown, and we cannot use it for demultiplexing. Because the actual sequence composition does not represent the true NTC for the corresponding PCR batch we did not remove those sequences detected in NTC6 from its controlled samples.”

The authors should add the results from all controls to the Supplementary Data files.

Response: Done, please see the Supplementary Data 1 and Data 2 (Extended excel files, sheet = “Controls_Data”). We have also indicated the information about PCR batches in both excel files (sheet = “PCR_Batches”).

Minor:

line 23: as currently written, it is unclear whether "one third of the vascular plant flora of China" refers to the entire Tibetan Plateau or the regions at ~3,600 m asl within the Plateau.

Response: We have changed the sentence to "the south-eastern Tibetan Plateau (Hengduan Mountains, Fig. 1a, red dotted line) which harbours one third of the vascular plant flora of China⁸". Please see lines 23-24.

8. Tang, Z., Wang, Z., Zheng, C. & Fang, J. Biodiversity in China's mountains. *Front. Ecol. Environ.* **4**, 347–352 (2006).

line 35: what does "habitat diversity" mean? Is it "total alpine habitat diversity"?

Response: We mean the diversity of habitats in general. We slightly rephrased this sentence.

New text line 35: It is uncertain whether their preferred habitats¹² or the diversity of habitat should be conserved to protect richness¹³.

12. Su, X., Han, W., Liu, G., Zhang, Y. & Lu, H. Substantial gaps between the protection of biodiversity hotspots in alpine grasslands and the effectiveness of protected areas on the Qinghai-Tibetan Plateau, China. *Agric. Ecosyst. Environ.* **278**, 15–23 (2019).

13. Wessely, J. et al. Habitat-based conservation strategies cannot compensate for climate-change-induced range loss. *Nat. Clim. Change* **7**, 823–827 (2017).

line 40-41: this sentence needs one or two citations.

Response: Done, we have cited the paper from Trivedi et al., 2008.

15. *Trivedi, M. R., Berry, P. M., Morecroft, M. D. & Dawson, T. P. Spatial scale affects bioclimate model projections of climate change impacts on mountain plants. Glob. Change Biol. 14, 1089–1103 (2008).*

line 52: move the lat-long and altitude data from the introduction to the methods (~line 165).

Response: Done, please see line 211.

line 65: specify vascular plant

Response: We have clarified it as terrestrial seed plants.

New text lines 67-68: “They were assigned to 218 terrestrial seed plant taxa with 100% best identity (Supplementary Data 1).”

line 72-73: what does "integrated catchment signal" mean?

Response: It means the vegetation signals within the lake catchment.

New text lines 85-86: “vegetation signals within the lake catchment than pollen as it is not impacted by upward plant material transport (Supplementary Figure 2)”

line 77-78: based on Supplementary Figure 2, this is not true - there is more emigration than immigration after 3.6 ka.

Response: We are sorry for this mistake. We have revised the calculation of turnover following the advice of Reviewer #1. The results indicate that immigration dominates turnover after 3.6 ka. Please see the Supplementary Figure 3: lines 663-665.

Supplementary Figure 3 | Proportional plant taxa turnover since 18 ka. Taxa gain is defined as the proportion of immigrants that appear in the lake catchment between the selected time periods, while taxa lost is the proportional disappearance of species.

line 82: state also that these 984 barcodes are likely to also include PCR artifacts.

Response: We have indicated it as below:

New text lines 104-105: “Sample processing-related errors (e.g. PCR and sequencing) may have slightly inflated taxa richness.”

lines 96, 133: "total habitable extent" and "simulated alpine habitat extent" initially confused me, as the Fig. 2 legend states "Alpine habitat area" and "simulated total habitat area". Recommend rephrasing the legend and perhaps explicitly referring to "black line" or "red area" in the text to clarify.

Response: Done. We have referred to them as black line (please see line 122) and red area (please see line 140).

line 104: what are "further processes"?

Response: The disturbances on unstable slopes due to ice-melting, as we indicate in text lines 131-132: “This might be attributed to disturbances on unstable slopes restricting vegetation establishment²⁷.”

27. Milner, A. M. et al. Glacier shrinkage driving global changes in downstream systems. *Proc. Natl. Acad. Sci.* **114**, 9770–9778 (2017).

line 109: why no show the Mg/Ca ratio throughout the record in Fig. 2?

Response: The Mg/Ca ratio is non-significant after adjusted degrees of freedom across all time intervals. Thus, we did not show it in our first version. Now, it is shown in Fig. 2j (please see line 585).

line 88-129: supplementary table 2 is referred to multiple times throughout this section. I suggest including the key results in a main text table, and keeping the full results in the supplement.

Response: Done. The correlation coefficients between total plant taxa richness and the predictor variables have been summarized in Table 1. The remaining results for single alpine families are kept in Supplementary Table 4.

New Table: lines 620-628

Table 1 | Summary of correlation coefficients between total plant taxa richness and the predictor variables

	18-10 ka					14-3.6 ka					10-0 ka				
	rho	adj p value	df	adj df	alpha level	rho	adj p value	df	adj df	alpha level	rho	adj p value	df	adj df	alpha level
Total habitat	0.257	0.524	34	10	.25	/					/				
Temperature	0.225	0.746	34	11	>.25	-0.728	1.32e-06	35	11	.01	-0.932	2.00e-15	33	11	.0005
Glacier's decay	-0.587	0.002	30	13	.025	/					/				
Alpine habitat	/					0.739	7.24e-07	35	11	.01	0.966	3.18e-20	33	11	.0005
Forested area	/					-0.739	7.24e-07	35	11	.01	0.966	3.18e-20	33	11	.0005
Mg/Ca ratio	0.381	0.159	34	11	.25	0.412	0.046	35	11	.25	0.164	1.000	33	11	>.25
Land-use	/					/					0.939	3.87e-16	33	11	.0005

rho: Spearman's Rank correlation coefficient
 adjusted p-value: "Bonferroni"
 df: degrees of freedom
 adjusted df: adjusted degrees of freedom
 alpha level: Directional alpha levels of critical values for Spearman's Rank correlation coefficient
 /: not a predictor variable in corresponding time transition
 predictor variable with alpha level <= 0.05 was in bold.

line 108-110: this result is non-significant (according to supplementary table 2), so ensure this is clearly stated.

Response: We have revised the text, please see:

New text lines 133-138: “The increases in pedogenic minerals (as indicated by sedimentary proxy Mg/Ca ratio from the same record²⁸, Fig. 2j) may have promoted the increase of richness of some alpine families (e.g. Polygonaceae, alpha level = 0.025; Ranunculaceae, alpha level = 0.025; Orobanchaceae, alpha level = 0.05, Supplementary Table 4), supporting the idea that soil development contributes to the coexistence of a large number of plant species²⁹. However, it is not the key driver for the total plant taxa richness (alpha level = 0.25, Table 1).”

New lines: 711-720

Supplementary Table 4 | Summary of correlation coefficients of taxa richness within the most common alpine families and selected predictor variables.

	18-10 ka					14-3.6 ka					10-0 ka				
	rho	adjusted p-value	df	adjusted df	alpha level	rho	adjusted p-value	df	adjusted df	alpha level	rho	adjusted p-value	df	adjusted df	alpha level
Polygonaceae															
Total habitat	0.693	1.13e-05	34	10	.025	/	/	/	/	/	/	/	/	/	/
Temperature	0.673	2.77e-05	34	11	.025	-0.276	0.394	35	13	.25	-0.882	1.05e-11	33	12	.0005
Glacier changes	-0.634	3.86e-04	30	13	.025	/	/	/	/	/	/	/	/	/	/
Alpine habitat	/	/	/	/	/	0.277	0.386	35	13	.25	0.908	1.92e-13	33	11	.0005
Mg/Ca ratio	0.706	6.17e-06	34	11	.025	0.190	1.000	35	12	>.25	0.117	1.000	33	11	>.25
Land-use	/	/	/	/	/	/	/	/	/	/	0.937	4.83e-16	33	11	.0005
Ranunculaceae															
Total habitat	0.605	3.67e-04	34	10	.05	/	/	/	/	/	/	/	/	/	/
Temperature	0.582	7.93e-04	34	11	.05	-0.731	1.13e-07	35	11	.01	-0.954	2.79e-18	33	11	.0005
Glacier changes	-0.265	0.573	30	13	.25	/	/	/	/	/	/	/	/	/	/
Alpine habitat	/	/	/	/	/	0.739	7.14e-07	35	11	.01	0.982	5.78e-25	33	11	.0005
Mg/Ca ratio	0.688	1.43e-05	34	11	.025	0.384	0.075	35	11	.25	0.184	1.000	33	11	>.25
Land-use	/	/	/	/	/	/	/	/	/	/	0.935	7.99e-16	33	11	.0005
Asteraceae															
Total habitat	0.283	0.379	34	10	.25	/	/	/	/	/	/	/	/	/	/
Temperature	0.261	0.498	34	11	.25	-0.428	0.33	35	11	.10	-0.032	1.000	33	13	>.25
Glacier changes	-0.704	2.76e-05	30	13	.005	/	/	/	/	/	/	/	/	/	/
Alpine habitat	/	/	/	/	/	0.431	0.031	35	11	.10	0.058	1.000	33	12	>.25
Mg/Ca ratio	0.354	0.137	34	11	.25	0.452	0.020	35	11	.10	0.325	0.227	33	13	.25
Land-use	/	/	/	/	/	/	/	/	/	/	-0.064	1.000	33	13	>.25
Orbanchaceae															
Total habitat	0.572	0.001	34	10	.05	/	/	/	/	/	/	/	/	/	/
Temperature	0.520	0.005	34	11	.10	-0.871	9.15e-12	35	11	.001	-0.896	1.32e-12	33	11	.0005
Glacier changes	-0.699	3.47e-05	30	12	.01	/	/	/	/	/	/	/	/	/	/
Alpine habitat	/	/	/	/	/	0.879	3.41e-12	35	11	.0005	0.918	3.67e-14	33	11	.0005
Mg/Ca ratio	0.585	0.001	34	10	.05	0.339	0.161	35	10	.25	0.007	1.000	33	11	>.25
Land-use	/	/	/	/	/	/	/	/	/	/	0.969	6.23e-21	33	11	.0005
Saxifragaceae															
Total habitat	0.002	1.000	34	11	>.25	/	/	/	/	/	/	/	/	/	/
Temperature	-0.031	1.000	34	11	>.25	-0.889	7.70e-13	35	11	.0005	-0.915	5.75e-14	33	11	.0005
Glacier changes	-0.496	0.016	30	13	.05	/	/	/	/	/	/	/	/	/	/
Alpine habitat	/	/	/	/	/	0.900	1.48e-13	35	11	.0005	0.939	3.23e-16	33	10	.0005
Mg/Ca ratio	0.154	1.000	34	11	>.25	0.496	0.007	35	10	.10	0.113	1.000	33	11	>.25
Land-use	/	/	/	/	/	/	/	/	/	/	0.949	1.73e-17	33	11	.0005

rho: Spearman's Rank correlation coefficient

adjusted p-value: "Bonferroni"

df: degrees of freedom

adjusted df: adjusted degrees of freedom

alpha level: Directional alpha levels of critical values for Spearman's Rank correlation coefficient

/: not a predictor variable in corresponding time transition

predictor variable with alpha level <= 0.05 was in bold.

28. Opitz, S., Zhang, C., Herzschuh, U. & Mischke, S. Climate variability on the south-eastern Tibetan Plateau since the Lateglacial based on a multiproxy approach from Lake Naleng – comparing pollen and non-pollen signals. *Quat. Sci. Rev.* **115**, 112–122 (2015).

29. Laliberté, E. et al. How does pedogenesis drive plant diversity? *Trends Ecol. Evol.* **28**, 331–340 (2013).

line 128: do the authors mean "a few taxa" rather than "selected individuals"?

Response: Yes, it has been changed to "a few taxa" (please see line 161).

line 133: change "confirms" to "is consistent with", as these experimental studies were not explicitly tested.

Response: Done (please see line 167).

line 135: for context and readability, state when habitat loss due to forest invasion occurred.

Response: Done.

New text lines 169-170: “alpine habitat loss due to forest invasion (10-0 ka, rho = 0.966, alpha level = 0.0005, Table 1).”

line 191: which DNA extraction protocol was used, and what was the mass of sediment input for extraction?

Response: New text lines 238-245: “Each extraction batch included nine samples (3-10 g sample⁻¹) and one extraction control, which was treated with a partially modified protocol of PowerMax[®] Soil DNA Isolation kit (Mo Bio Laboratories, Inc. USA). The isolation of DNA was first processed by loading 15 mL PowerBead solution, 1.2 mL C1 buffer, 0.8 mg proteinase K (VWR International), 0.5 mL 1 M dithiothreitol (VWR International) and samples into PowerBead tubes. Then, all tubes were vortexed in 10 min and incubated at 56°C in a rocking shaker overnight under the aluminium foil protection. The subsequent extraction steps follow the kit manufacturer’s instructions and were completed on the second day.”

line 195-202: what was the final reaction volume?

Response: A total of 25 µL per PCR reaction (please see line 250).

line 207: presumably "ancient DNA separated" means "ancient DNA lab, physically separated"?

Response: Yes.

New text lines 260-262: “PCR set-ups were conducted under a dedicated UV working station in the detached ancient DNA laboratory physically separated from the workplace of Post-PCR where we did the thermal cycling, purification and pooling.

line 207: clearly state that each PCR batch was replicated once for two PCR replicates per sample. line 208: define "NTC".

Response: Done. The definition of “NTC” (no template control) in line 258.

New text lines 262-264: “Each PCR batch was replicated until obtaining two positive PCR replicates for each lake sediment sample when the associated controls were negative.”

line 207-208: what does "positive sample PCR products" mean? Only those with visible bands on a gel?

Response: New text lines 264-268: "A qualified positive PCR product was considered only if it matched two conditions: (1) the gene band is evidently longer than that of negative controls; (2) the brightest staining is in the 100-200 bp range. Specifically, the thin/blurry products below 50 bp in corresponding controls are primer dimers and not expected PCR products. The gene band was found with 2% agarose gel electrophoresis."

line 223: add additional citations for ArctBorBryo (Willerslev et al. 2014, Nature; Soininen et al. 2015, PLoS One).

Response: Done (please see line 289).

48. *Sønstebo, J. H. et al. Using next-generation sequencing for molecular reconstruction of past Arctic vegetation and climate. Mol. Ecol. Resour. 10, 1009–1018 (2010).*

49. *Willerslev, E. et al. Fifty thousand years of Arctic vegetation and megafaunal diet. Nature 506, 47–51 (2014).*

50. *Soininen, E. M. et al. Highly Overlapping Winter Diet in Two Sympatric Lemming Species Revealed by DNA Metabarcoding. PLoS ONE 10, e0115335 (2015).*

line 241: state that one sample (ESL024) was dropped, due to no taxa detected.

Response: Done.

New text lines 305-306: "The sample ESL024 was excluded from further statistical analysis because no reads were obtained."

line 272: rephrase "sample size" as "sequencing depth" or similar.

Response: We have corrected it to "read counts" (please see line 354).

line 282: remove "Pleistocene-Holocene transition" as this is at 11.7 ka, not 18-10 ka.

Response: Done.

line 297-299: what about human/grazing indicators?

Response: Supplementary Table 5 indicates that the human/grazing indicators explained 86.31% of the deviance of total plant taxa richness, which is less than that of alpine habitat area (96.04%). In addition, the alpine habitat area is more important even considering both variables in a GLM model (alpine habitat area: 9.73, human/grazing: 2.16). The alpine habitat area explained most of the deviance and more important for single alpine families as well. Thus, we

predicted the total plant taxa richness and within-family richness using the relationship between alpine habitat area and plant taxa richness.

New text lines 407-410: “We predicted the total richness and within-family richness for 2050-2300 CE in 50-year time steps using ‘*glm.predict()*’ based on the most important predictor variable (alpine habitat area) in the GLM models. The variable importance was calculated using “*varImp()*”.”

line 464: are the CONISS clusters based on taxonomic composition or relative read abundance?
Response: Base on the relative read abundance (please see line 599).

line 486: specify this is "relative read abundance"

Response: Done (please see line 657).

line 502: invert x-axis of Supplementary Figure 4 to be consistent with other plots.

Response: Done (please see line 676).

line 512: this could be a single plot with different coloured lines, as the current plots are misleading due to the different y-axis ranges.

Response: We re-plotted them with the same y-axis ranges (please see line 673).

Supplementary Table 1: divide the "identification ability" values by 100.

Response: Following the comment of Reviewer #3, this column ("identification ability") has been deleted.

Supplementary Table 2: replace "non-sig" with the correct alpha level. Suggest highlighting rows with a significant result in bold to enhance readability.

Response: Done, please see new Table 1 (lines 620-628) and Supplementary Table 4 (lines 711-720).

line 530: should "sFamily" be "sRichness"? What does "Aa" mean?

Response: Yes. We have corrected it (please see line 724) and also added the annotation for "Aa" which means “alpine habitat area” (please see line 726).

Typos/grammer:

line 6: instead of two abbreviations, perhaps just say "18,000 years".

Response: Done (please see line 6).

line 95 and throughout: replace "insignificant" with "non-significant".

Response: Done (please see line 121).

line 137: correct spelling of "conservation".

Response: Done (please see line 194).

line 167-168: change "glacier activities" to "glacial activity".

Response: Done (please see line 214).

line 208: change "having a" to "including the".

Response: New text without both words (please see line 264).

line 233: correct "Table " to "Data".

Response: Done (please see line 299).

lines 253, 256, etc: correct citations.

Response: Done (please see line 322 and line 325).

line 288: change "check significance" to "check the significance".

Response: Done (please see line 352).

line 477: change "are" to "were"

Response: Done (please see line 619).

line 451: delete "areal".

Response: Done (please see line 584).

line 508: delete "in" and change "band across the" to "bands across".

Response: Done (please see line 681).

The responses are in blue. The revisions are marked in red in the revised manuscript. The comments were separated into several parts and responded to point by point.

Reviewer #3 (Remarks to the Author):

This manuscript entitled ‘Alpine habitat loss threatens the future plant diversity of the Tibetan Plateau’ applied a time-series approach to infer past richness from sedimentary ancient DNA of one core from a catchment area in the south eastern Tibetan Plateau over the last 18 ka (cal ka BP). Based on the established relationship between plant richness and environmental changes, especially changes of the ‘alpine habitats’, the authors concluded that a simulated alpine habitat loss in a warmer future could cause a 41% decrease in plant richness at the study site over the next 250 years. The MS was well written but has some unclear statements and uncertain speculations. This work well described the historical changes in plant richness and vegetation composition with a relatively high resolution. The sedimentary ancient DNA method showed an advanced and practical way to infer such historical changes across a long time span. However, the results contain a lot of speculations due to scale issue, which makes conclusions not so convincing. To date, both field observations and model predictions in either European Alps or Asia mountains (also suggested by Liang et al 2018, J BIOGEOGR, the reference 5) indicated that upward shift of alpine plant species would result in high species richness in the context of climate warming. Though the authors drew conclusions challenging these findings, I did not see any superior approaches of this study compared with traditional methods, such as SDMs, to make the story plausible. The major problem is that the authors imposed coarse scale relationship established between species richness and environmental change from history on predicting fine scale biodiversity change in the future. In other words, the spatial-temporal scale throughout the analysis was not consistent at several dimensions. Here I listed several issues that the authors did not clearly address in the MS.

Response: We thank reviewer#3 for his/her comments.

Major comments

(1) Macrorefugia vs microrefugia. I think alpine habitat extent in this study somehow refers to macrorefugia. If this is the case, the authors should refer to the paper from THEOFANIA et al 2014 GCB, from which it clearly showed the importance of supporting functions of microrefugia when forecasting the fate of alpine plant species under climate change. In this work, I did not see any concerns on this issue.

Response: Thank you for pointing us to the macrorefugia vs. microrefugia debate. We agree with that the microrefugia should be considered for predicting the fate of species under climate change (also highlighted by Theofania et al 2014, GCB). We also agree that our simulation approach has several biases. However, in contrast to traditional SDMs that ignore microrefugia, our approach implicitly deals with the microrefugium vs. macrorefugium problem because it relates the richness of an entire catchment (including all microrefugia) to climate. Hence, predictions for the very same catchment implicitly reflect the richness change in the entire catchment. We highlight this as an advantage of our approach.

New text lines 189-193: “For example, in contrast to traditional SDMs (species distribution models) that ignore microrefugia, our approach implicitly deals with the microrefugium vs. macrorefugium problem because it relates the plant taxa richness of the entire catchment (including all microrefugia) to climate. Hence, predictions for the very same catchment implicitly reflect the plant taxa richness change of the entire lake catchment.”

(2) To what extent, the information derived from only one core could be extrapolated to a large scale? Alpine ecosystem is featured by high heterogenous landscape and rugged terrain. Is it confident that one core could capture the spatial environmental variations (altitudinal and horizontal) of this alpine ecosystem? Would it benefit from more cores since Naleng lake is not the only lake in this study area as indicated from your previous work (reference 34)?

Response: We agree that more records would be good to confirm our observed diversity pattern. However, our sedaDNA data are supported by the pollen signal, i.e. our pollen-based vegetation change agrees with other pollen records from the Tibetan Plateau (reviewed by Chen et al. 2020; Hou et al. 2017). Thus, we are confident that we investigated a very “typical alpine lake” that archived the main signal of the south-eastern Tibetan alpine ecosystem. Furthermore, we assume, in accordance with modern studies, that the lake system integrates the signal over the entire catchment, in contrast, for example to vegetation plot studies. Hence, if at all, only differences between catchments would be problematic. However, the elevation of Lake Naleng catchment (4,400–4,800 m) covers the most typical elevation of the Hengduan Mountains, as can be seen in the elevation distribution in Fig. 1a, b (larger area of Hengduan Mountains from ~ 4,200 to ~ 4,900 m a.s.l.). Accordingly, we assume that the inferred biodiversity pattern from Naleng is characteristic of alpine areas in the Hengduan Mountains.

New text lines 85-90: “The *seadNA* better captures the vegetation signals within the lake catchment than pollen as it is not impacted by upward plant material transport (Supplementary Figure 2) and records more taxa at higher taxonomic resolution than pollen spectra (Supplementary Table 3). Accordingly, and because the lake catchment covers the most common elevations in the Hengduan Mountains (~ 4,200 - ~ 4,900 m a.s.l., Fig. 1a, b), we conclude that Lake Naleng archived the main signal of the south-eastern Tibetan alpine ecosystem.”

Hou, G., Yang, P., Cao, G., Chongyi, E. & Wang, Q. Vegetation evolution and human expansion on the Qinghai–Tibet Plateau since the Last Deglaciation. Quat. Int. 430, 82–93 (2017).

Chen, F. et al. Climate change, vegetation history, and landscape responses on the Tibetan Plateau during the Holocene: A comprehensive review. Quat. Sci. Rev. 243, 106444 (2020).

(3) The established relationship between species richness and historical environmental change of the past 18 ka could be used to predict future change of the next 250 years? You divided the past 18 ka into several stages, the time span of each stage far exceeds 250 years. Though the authors included simulation of the future change of alpine habitat extent, it seems the constructed relationship at coarse temporal scale was imposed to predict future changes at fine temporal scale, which is very difficult to understand.

Response: To clarify, we only used the richness-alpine habitat area relationship for 10–0 ka as an analogue to predict the richness change in the future. This period was selected as it covers the warmest and most modern phase of the record. Moreover, the correlation between total plant taxa richness and alpine habitat is highest in this time interval. The temporal resolution of the correlated time-series was about 250 years.

We estimated the alpine habitat area in the past (based on the climate proxy-data and constrained by the modern treeline position) and for the future catchment (based on climate forecast for a RCP4.5; in Supplementary Figure 5). Using GLM modelling (see Supplementary Table 5), we identified a strong link between taxa richness (also for some alpine families) and the available alpine habitat area in the past, which was then applied to the prediction of richness in future.

We agree with you that our predictions involve some uncertainties because treeline response is slow on a decadal time-scale but is in “quasi-equilibrium” on a multi-centennial time-scale as described in the GLM model. We now report our prediction on only the multi-centennial time-scale, i.e. for 2300 CE.

Because the current warming exceeds previous warming rates our model may overestimate the vegetation responsiveness. Unfortunately, treeline response in the Hengduan

Mountains has not yet been investigated and even temperature records are scarce for such high elevations. So, our predictions cannot be validated by recent observations and we refrain from that in the manuscript. The nearest sites investigated (ca. 250 km away) with *Picea* as the treeline-forming tree showed a rise of about 70 m in 100 yr indicating that *Picea* treelines are currently rising under a warming rate of about 1 °C in the past 100 yr (Liang et al., 2016).

We agree that our calculations cannot be considered as reliable predictions but rather as a potential scenario by analogy to the past and under consideration of the limitations of the approach we applied. We indicate this in new text version.

New text lines 181-186: “Investigations of forest changes are lacking for our study area. However, the upper alpine *Picea* forest line has risen by about 70 m during the last 100 years in other mountain ranges of the south-eastern Tibetan Plateau, indicating its sensitivity to warming³⁶. Even though our data are insufficient to make a reliable prediction, our simulated taxa loss in relation to shrinking alpine habitat extent should be treated as a potential scenario by analogy to the past.”

New text lines 400-405: “A generalized linear model (GLM) was built using ‘*glm(family = gaussian)*’ for total plant taxa richness and those families that are significantly related to the predictor variables (alpine habitat area and land-use indicator) during 10-0 ka (Supplementary Code 4). This period was selected as it covers the warmest and most modern phase of the record. Moreover, the correlation between total plant taxa richness and alpine habitat is highest in this time interval. The temporal resolution of the correlated time-series was about 250 years.”

36. Liang, E. et al. *Species interactions slow warming-induced upward shifts of treelines on the Tibetan Plateau. Proc. Natl. Acad. Sci. 113, 4380–4385 (2016).*

(4) The simulation of alpine habitat extent. First, I find it very difficult to interpret how climate changed during the simulation period. Why climate data of a meteorological station 80 km away at 50 m a.s.l. was applied to calculate the temperature of a mountainous area with elevation ranging from 1000 m to 6000 m a.s.l.?

Response: We did not use climate data from Chengdu climate station, which was erroneously stated in the former manuscript text and is now corrected. Instead, we use a temperature lapse rate (Li et al., 2013) to translate the temperature anomaly (derived from proxy-data or prediction) to shift the elevation area in our catchment. The alpine habitat area is then calculated as the area above the treeline and below the (dynamic) glacier cover.

For example, as Supplementary Figure 7 (please see lines 676) shows, area between an elevation of 4,600–5,100 m a.s.l. were counted at 16 ka because the temperature was colder than present. By contrast, the area between 4,000–4,800 m a.s.l. were counted at 6 ka because the temperature anomaly was warmer than present.

New text lines 328-343: “We then modelled the available habitable area within the lake catchment back-in-time (Supplementary Figure 7) according to the following steps: (1) delineate the catchment using the global 1-arcsecond (90-m) SRTM digital elevation model and downscale to 30-m resolution for simulation; (2) combine the two reconstructed past temperature records^{56,57}; (3) calculate the relative elevation of each pixel based on the temperature lapse rate of $0.55^{\circ}\text{C}/100\text{m}^{64}$ for the catchment over the past 18 ka for each step of 500 years by the relative temperature change from the constructed temperature series in (2); (4) calculate the elevational range of the catchment over the past 18 ka under the effect of simulated glacier cover; (5) group the elevation values per 100-m elevational band ranging from 1000 m a.s.l. to 6000 m a.s.l. under the effect of simulated glacier cover and sum the pixels in all elevational bands as total habitable area (Fig. 2h, black outline); (6) sum up the number of pixels above the modern treeline ($\sim 4,400$ m a.s.l.) to obtain the alpine habitat area (Fig. 2h, red polygon); and (7) compute the elevational range above the modern treeline in the catchment (2050–2300 CE) under the projections of temperature according to RCP 4.5 emissions scenario (source: <http://svn.zmaw.de/svn/cosmos/branches/releases/mpi-esm-cmip5/src/mod>) for indicating the loss of alpine habitat under the ongoing climate warming (Supplementary Figure 5).”

56. Shakun, J. D. et al. Global warming preceded by increasing carbon dioxide concentrations during the last deglaciation. *Nature* **484**, 49–54 (2012).

57. Marcott, S. A., Shakun, J. D., Clark, P. U. & Mix, A. C. A Reconstruction of Regional and Global Temperature for the Past 11,300 Years. *Science* **339**, 1198–1201 (2013).

64. Li, X. et al. Near-surface air temperature lapse rates in the mainland China during 1962–2011. *J. Geophys. Res. Atmospheres* **118**, 7505–7515 (2013).

Second, how treeline changed during the simulation period, which process tightly linked to the area above treeline?

Response: As pointed out in the preceding response, we assessed the treeline change during the simulation period by determining the modern elevational position and linking this with the catchment elevation information at a certain time point.

We extended the text.

New text lines 343-347: “The modern treeline was calculated based on 49 current treeline points taken from publications and high-resolution satellite images in GoogleEarth™ (Supplementary Figure 8 and Supplementary Table 6). It should be noted that uncertainties in the simulation may arise from species interactions and potentially lagging treeline response to climate warming³⁶.”

36. Liang, E. et al. Species interactions slow warming-induced upward shifts of treelines on the Tibetan Plateau. *Proc. Natl. Acad. Sci.* **113**, 4380–4385 (2016).

Thirdly, according to Supplementary Figure 5, does it mean by the year 2300, alpine habitats above treeline was almost lost? due to forest invasion? Again, since it is not clear how treeline responds to the environmental change in this area, such result needs more evidences. Finally, it is also very uncertain how species richness followed the habitat change in the next 250 years. For example, if alpine habitat extent decreased by 10%, how species richness responded to such change? It seems there were too many speculations from this part.

Response: As indicated above, our approach does not specifically consider time-lagged response, but it assumes a “quasi-equilibrium” on a multi-centennial time-scale between temperature and vegetation change. We no longer present our simulation results with a decadal resolution but just for 2300 CE and indicate that it should be considered as a potential scenario rather than a reliable prediction of the diversity dynamics.

We refer the reviewer to Fig. 3a in which the taxa richness can be checked for changes in alpine habitat area. For example, a 10% decrease from 100 km² would cause a drop of ~5 taxa.

Fig. 3 Predicted total plant taxa richness under 2.5°C climate warming between 2050 and 2300 based on the inferred past relationship between total plant taxa richness and alpine habitat area. a, The relationship between total plant richness and alpine habitat area was established by a generalized linear model (Methods).

New text line 11: we have changed the “41%” to “substantive”.

New text line 176: we have changed the “41%” to “pronounced”.

New text lines 181-186: “Investigations of forest changes are lacking for our study area. However, the upper alpine *Picea* forest line has risen by about 70 m during the last 100 years in other mountain ranges of the south-eastern Tibetan Plateau, indicating its sensitivity to warming³⁶. Even though our data are insufficient to make a reliable prediction, our simulated taxa loss in relation to shrinking alpine habitat extent should be treated as a potential scenario by analogy to the past.

New text lines 187-189: “Even if our simulation overestimates the loss of alpine habitats (because the high current warming rates are unprecedented during the Holocene) it may complement other modelling approaches.”

36. Liang, E. et al. *Species interactions slow warming-induced upward shifts of treelines on the Tibetan Plateau. Proc. Natl. Acad. Sci.* **113**, 4380–4385 (2016).

(5) The number of plant species identified by sedaDNA was obviously much lower than that of current species pool in this study area. Despite of biological interaction and evolutionary adaption of alpine plant species, is the identified number of historical plant species large enough to predict future biodiversity change?

Response: We agree that our approach underestimates the total plant taxa richness, mainly because of limitations in the current taxonomic reference database and due to the limited specificity of the marker used. That said, we are confident that the sedaDNA approach is a powerful tool for tracking the long-term relative changes of taxa richness, as we found that our observed temporal richness pattern is not sensitive to the dataset used (please see the new Supplementary Table 1) and also because species-level richness is rather similar to richness at lower taxonomic levels (that can be resolved by the marker).

New text lines 100-104: “Furthermore, the similar temporal pattern of total plant taxa richness was obtained when analysing the data before taxonomic assignment and the data containing all

terrestrial seed plant sequences (Methods), as indicated by the highly significant correlations of these time-series with total plant taxa richness (Supplementary Table 1).”

New Supplementary Table: lines 688-700

Supplementary Table 1 | The correlation of plant taxa richness based on different datasets indicates that the temporal variations of total plant taxa richness were accurately reflected by the data with best identity 1 in this study

	rho	p-value	df
Bestid 1 vs. Non-ecotag data	0.768	5.38e-15	69
Bestid 1 vs. terSeq data	0.830	3.52e-19	69
Bestid 1 vs. Bestid 0.95	0.888	6.46e-25	69
Bestid 1 vs. Single data	0.951	7.74e-37	69

Bestid 1: after further sequence filtering, collected those terrestrial seed plant sequences that having best identity value of 1 and present in two PCRs at least.

Bestid 0.95: after further sequence filtering, collected those terrestrial seed plant sequences that having best identity value ≥ 0.95 and present in two PCRs at least.

terSeq: the data containing all terrestrial seed plant sequences without further sequence filtering.

Non-ecotag: the data before taxonomic assignment

Single data: based on data Bestid 1, only using deeply sequenced PCR replicate for each lake-sediment sample.

rho: Spearman's Rank correlation coefficient

df: degrees of freedom

For detail information, please see Methods.

New text lines 105-111: “However, we assume that we rather underestimate richness because of non-specificity of the marker and non-completeness of the reference database, which likely means that the *seadaDNA* detected taxa number is lower than the absolute taxa number recorded in the flora. Additionally, species-rich families²⁵ in the flora including Asteraceae, Saxifragaceae, and Orobanchaceae have highest richness in our record. Thus, plant taxa richness (total plant taxa richness and taxa richness within dominant alpine families) can be regarded as a semi-quantitative proxy of taxa richness in our study.”

25. Yu, H. et al. *Contrasting Floristic Diversity of the Hengduan Mountains, the Himalayas and the Qinghai-Tibet Plateau Sensu Stricto in China*. *Front. Ecol. Evol.* **8**, (2020).

Minor comments

(1) The authors argued methodological limitations of the traditional space-for-time approaches. However, I did not find any comparison or justification on how superior their methodology is.

Response: The advantages of a time-series approach over the traditional space-for-time approach are that the sampled site is constant (i.e. normalizing for the “sampling effect”), that sampling elevation always represents the same portion of the investigated mountain range (i.e.

normalizing for the “area effect”) and that it is always placed at the same relative elevation (i.e. normalizing for the “mid-domain” effect). Please see lines 41-45. We have added:

New text lines 45-48: “Hence, such an approach can well reflect the temporal biodiversity-environmental relationship and as such maximize the effects of relevant variables when predicting biodiversity change over time¹⁶.”

New text lines 189-193: “For example, in contrast to traditional SDMs (species distribution models) that ignore microrefugia, our approach implicitly deals with the microrefugium vs. macrorefugium problem because it relates the plant taxa richness of the entire catchment (including all microrefugia) to climate. Hence, predictions for the very same catchment implicitly reflect the plant taxa richness change in the entire lake catchment.”

16. Gavin, D. G. et al. *Climate refugia: joint inference from fossil records, species distribution models and phylogeography*. *New Phytol.* **204**, 37–54 (2014).

(2) Line 156, what do you mean by protect? I did not see any practical suggestions to ‘protect’ the upper alpine habitat.

Response: We decided against suggesting detailed protection measures as this is outside the scope of our investigation.

(3) Line 246, how the numerical ice flow model GC2D worked? Needs descriptions. If it is only about how glacier retreated during the past 18 ka, then I doubt its application to simulate changes of alpine habitats since plant establishment (e.g., treeline species) after glacier retreat related to seed dispersal, available microhabitats, and climate, etc.

Response: We have indicated that the ice-flow model only simulates the past flow of ice and any concurrent advance and retreat (please see Line 334). It does not consider any delayed taxa invasion response. However, the modern study shows that pioneer trees (*Hippophae tibetana*, *Larix griffithii* and *Pinus wallichiana*) rapidly colonize (4–11 years) the ice-free area after glacier retreat on the Tibetan Plateau (Zhu et al., 2019). Accordingly, it is reasonable to link habitable area changes (as derived from modelled glacier retreat) to taxa richness on centennial and millennial scales.

New text lines 326-327: “To clarify, the ice-flow model only simulated the past flow of ice and any concurrent advance and retreat.”

Zhu et al. 2019. Trees record changes of the temperate glaciers on the Tibetan Plateau: Potential and uncertainty. Global and Planetary Change, 173, pp.15-23.

(4) Line 445, better if location of the catchment b) to be shown in a)

Response: We tried it, but it is too small to see. Thus, we used the same icon as shown in Fig. 1b to indicate the location of the lake sediment core.

(5) Line 466, the landscape change is true or just speculated, or from the GC2D? If it is not true, I would prefer a table instead.

Response: The elevation in the sketch of the catchment is true. We decided to stay with the sketches about richness change and impact of potential drivers to allow non-specialists to infer the main points from our work at a glance. Furthermore, we have added Table 1 to show the main correlations.

New Table 1: lines 620-628

Table 1 Summary of correlation coefficients between total plant taxa richness and the predictor variables

	18-10 ka					14-3.6 ka					10-0 ka				
	rho	adj p-value	df	adj df	alpha level	rho	adj p-value	df	adj df	alpha level	rho	adj p-value	df	adj df	alpha level
Total habitat	0.257	0.524	34	10	.25	/					/				
Temperature	0.225	0.746	34	11	>.25	-0.728	1.32e-06	35	11	.01	-0.932	2.00e-15	33	11	.0005
Glacier's decay	-0.587	0.002	30	13	.025	/					/				
Alpine habitat	/					0.739	7.24e-07	35	11	.01	0.966	3.18e-20	33	11	.0005
Forested area	/					-0.739	7.24e-07	35	11	.01	0.966	3.18e-20	33	11	.0005
Mg/Ca ratio	0.381	0.088	34	11	.25	0.412	0.046	35	11	.25	0.164	1.000	33	11	>.25
Land-use	/					/					0.939	3.87e-16	33	11	.0005

rho: Spearman's Rank correlation coefficient
adjusted p-value: "Bonferroni"
df: degrees of freedom
adjusted df: adjusted degrees of freedom
alpha level: Directional alpha levels of critical values for Spearman's Rank correlation coefficient
/: not a predictor variable in corresponding time transition
predictor variable with alpha level <= 0.05 was in bold.

(6) Line 479, Figure 4. No surprising extremely low species richness if there is no alpine habitat existed. Again, how changes of alpine habitat extent led to the species richness change is not clear.

Response: We do not agree with this comment. Looking at the modern taxa distribution that peaks at 3500 m a.s.l. one would expect an increase in taxa richness with a rising treeline, as predicted by current SDMs in response to climate warming (e.g. Liang et al., 2018, also see Fig. 1a). Our model, in contrast, predicts a taxa loss in response to future warming and related alpine habitat loss. This highlights the advantages of our applied time-for-time approach.

Fig. 1a, Area-elevation relationship (grey bars), elevational species richness distribution (red dotted line) and forest zone (blue dashed line) are shown lower-right panel.

Liang et al. 2018. Shifts in plant distributions in response to climate warming in a biodiversity hotspot, the Hengduan Mountains. Journal of Biogeography, 45(6), pp.1334-1344.

(7) Line 515, commonly, sedaDNA obtained high taxonomic resolution but identified less taxa than pollen, which is contracted to Supplementary Table 1. It won't hurt if authors explained this a little bit.

Response: We revised the text. The *sedaDNA* identified more taxa than pollen. For example, 218 unique taxa were obtained from 71 *sedaDNA* samples, but 191 pollen samples only provided 152 unique taxa.

New lines: 708-710

Supplementary Table 3 | Identified taxa and taxonomic resolution. We compared number of taxa and their taxonomic resolution of *sedaDNA* data with pollen data from the same sediment core. Taxonomic resolution is indicated by the percentage of taxa within each proxy.

	No. of Samples	No. of Taxa	Taxonomic resolution (%)						
			Subspecies	Species	Genus	Subtribe	Tribe	Subfamily	Family
sedaDNA	71	218	0.46	31.19	39.45	2.75	5.50	8.72	11.93
Pollen	191	152	0	23.03	63.82	0	0	1.32	11.84

Reviewers' Comments:

Reviewer #1:

Remarks to the Author:

I am satisfied with the revision made on the manuscript related to my part of the review.

Reviewer #2:

Remarks to the Author:

I thank the authors for doing an excellent job of thoroughly addressing my concerns - the manuscript is greatly improved.

Barring one minor clarification, I recommend that the manuscript be accepted for publication in Nature Communications:

L. 372-373: uncertain what '0.25%' is referring to - is this minimum relative read abundance?

Peter Heintzman

Reviewer #3:

Remarks to the Author:

Please see attached.

- (1) Macrorefugia vs microrefugia. I think alpine habitat extent in this study somehow refers to macrorefugia. If this is the case, the authors should refer to the paper from THEOFANIA et al 2014 GCB, from which it clearly showed the importance of supporting functions of microrefugia when forecasting the fate of alpine plant species under climate change. In this work, I did not see any concerns on this issue.

Authors' Response: Thank you for pointing us to the macrorefugia vs. microrefugia debate. We agree with that the microrefugia should be considered for predicting the fate of species under climate change (also highlighted by Theofania et al 2014, GCB). We also agree that our simulation approach has several biases. However, in contrast to traditional SDMs that ignore microrefugia, our approach implicitly deals with the microrefugium vs. macrorefugium problem because it relates the richness of an entire catchment (including all microrefugia) to climate. Hence, predictions for the very same catchment implicitly reflect the richness change in the entire catchment. We highlight this as an advantage of our approach.

New comment: I doubt that the catchment in this study covers all microrefugia in alpine ecosystems. Microrefugia is actually a combination of several biotic and abiotic factors, which itself is a sophisticated term and hard to quantify. Anyway, a misunderstanding about SDMs should be avoided. I would not say SDMs ignore microrefugia. On the contrary, the SDMs consider hundreds or thousands sampling points which highly related to microhabitat information. Perhaps it is good to read some papers like 'Microclimate and demography interact to shape stable population dynamics across the range of an alpine plant', 'Extinction debt of high-mountain plants under twenty-first-century climate change'. Even if the catchment covered all microrefugia, one sediment core did not record any information relating to microrefugia, which is exactly the point that I argue you did not take microrefugia into account to predict future biodiversity change. I totally agree that the method used in this study is a new and interesting way to understand alpine biodiversity change. However, since the authors claimed in the abstract that their findings challenge the idea that future warming and treeline rise is expected to cause an increase of plant diversity in present alpine habitats, I think they should compare their prediction results to those from SDMs (as far as I know, there are also some predictions based on SDMs for the next century) and discuss advantages or limitations compared with SDMs. Otherwise, it would be very hard for audiences to understand the superiority of their method.

- (2) To what extent, the information derived from only one core could be extrapolated to a large scale? Alpine ecosystem is featured by high heterogenous landscape and rugged terrain. Is it confident that one core could capture the spatial environmental variations (altitudinal and horizontal) of this alpine ecosystem? Would it benefit from more cores since Naleng lake is not the only lake in this study area as indicated from your previous work (reference 34)?

Authors' Response: We agree that more records would be good to confirm our observed diversity

pattern. However, our sedaDNA data are supported by the pollen signal, i.e. our pollen-based vegetation change agrees with other pollen records from the Tibetan Plateau (reviewed by Chen et al. 2020; Hou et al. 2017). Thus, we are confident that we investigated a very “typical alpine lake” that archived the main signal of the south-eastern Tibetan alpine ecosystem. Furthermore, we assume, in accordance with modern studies, that the lake system integrates the signal over the entire catchment, in contrast, for example to vegetation plot studies. Hence, if at all, only differences between catchments would be problematic. However, the elevation of Lake Naleng catchment (4,400–4,800 m) covers the most typical elevation of the Hengduan Mountains, as can be seen in the elevation distribution in Fig. 1a, b (larger area of Hengduan Mountains from ~ 4,200 to ~ 4,900 m a.s.l.). Accordingly, we assume that the inferred biodiversity pattern from Naleng is characteristic of alpine areas in the Hengduan Mountains.

New comment: It is not because the elevation range (4,400–4,800 m) makes Lake Naleng catchment a very typical lake to conduct such study, as a good sediment sample core actually matters. My argument is not how ‘typical’ the catchment is, which you could not quantify, but how one core information provides comprehensive information of vegetation in rugged terrain of vast Tibet plateau. For example, if there were two or three catchments separated by a mountain ridge, which is very common in alpine ecosystems, and you obtained some cores from the catchments, then the differences among the cores would tell how confidence the derived information is, especially at different time period.

(3) The established relationship between species richness and historical environmental change of the past 18 ka could be used to predict future change of the next 250 years? You divided the past 18 ka into several stages, the time span of each stage far exceeds 250 years. Though the authors included simulation of the future change of alpine habitat extent, it seems the constructed relationship at coarse temporal scale was imposed to predict future changes at fine temporal scale, which is very difficult to understand.

Authors’ Response: To clarify, we only used the richness-alpine habitat area relationship for 10 – 0 ka as an analogue to predict the richness change in the future. This period was selected as it covers the warmest and most modern phase of the record. Moreover, the correlation between total plant taxa richness and alpine habitat is highest in this time interval. The temporal resolution of the correlated time-series was about 250 years. We estimated the alpine habitat area in the past (based on the climate proxy-data and constrained by the modern treeline position) and for the future catchment (based on climate forecast for a RCP4.5; in Supplementary Figure 5). Using GLM modelling (see Supplementary Table 5), we identified a strong link between taxa richness (also for some alpine families) and the available alpine habitat area in the past, which was then applied to the prediction of richness in future. We agree with you that our predictions involve some uncertainties because treeline response is slow on a decadal time-scale but is in “quasi-equilibrium” on a multi-centennial time-scale as described in the GLM model. We now report our prediction on only the multicentennial time-scale, i.e. for 2300 CE. Because the current warming exceeds previous

warming rates our model may overestimate the vegetation responsiveness. Unfortunately, treeline response in the Hengduan Mountains has not yet been investigated and even temperature records are scarce for such high elevations. So, our predictions cannot be validated by recent observations and we refrain from that in the manuscript. The nearest sites investigated (ca. 250 km away) with *Picea* as the treeline-forming tree showed a rise of about 70 m in 100 yr indicating that *Picea* treelines are currently rising under a warming rate of about 1 ° C in the past 100 yr (Liang et al., 2016). We agree that our calculations cannot be considered as reliable predictions but rather as a potential scenario by analogy to the past and under consideration of the limitations of the approach we applied. We indicate this in new text version.

New comment:

Treeline position highly affects the changes of alpine habitat extent defined in this study.

However, if you can not prove that you obtained reliable dynamic changes of treeline, how confident the predictions of alpine habitat extent are?

Again, the relationship between total plant taxa richness and alpine habitat in the time period of 10 – 0 ka is too rough to predict future biodiversity change since some important processes are ignored, such as snow cover change, upward shift of low land plant species, species interaction... etc, which are widely believed as key processes affecting species richness in alpine ecosystems. Moreover, the prediction mixed the processes of trees and low stature plants above treeline, which also could result in unreliable predictions since responses of, for example, trees and herbs are obviously different.

(4) The simulation of alpine habitat extent. First, I find it very difficult to interpret how climate changed during the simulation period. Why climate data of a meteorological station 80 km away at 50 m a.s.l. was applied to calculate the temperature of a mountainous area with elevation ranging from 1000 m to 6000 m a.s.l.? Second, how treeline changed during the simulation period, which process tightly linked to the area above treeline? Thirdly, according to Supplementary Figure 5, does it mean by the year 2300, alpine habitats above treeline was almost lost? due to forest invasion? Again, since it is not clear how treeline responds to the environmental change in this area, such result needs more evidences. Finally, it is also very uncertain how species richness followed the habitat change in the next 250 years. For example, if alpine habitat extent decreased by 10%, how species richness responded to such change? It seems there were too many speculations from this part.

Authors' Response: We did not use climate data from Chengdu climate station, which was erroneously stated in the former manuscript text and is now corrected. Instead, we use a temperature lapse rate (Li et al., 2013) to translate the temperature anomaly (derived from proxy-data or prediction) to shift the elevation area in our catchment. The alpine habitat area is then calculated as the area above the treeline and below the (dynamic) glacier cover. As indicated above, our approach does not specifically consider time-lagged response, but it assumes a “quasi-equilibrium” on a multi-centennial time-scale between temperature and vegetation change. We no longer present our simulation results with a decadal resolution but

just for 2300 CE and indicate that it should be considered as a potential scenario rather than a reliable prediction of the diversity dynamics.

New comment: Temperature lapse rate might be a good solution if there's no available adjacent climate station. However, climate data from Chengdu station, a big city with high urbanization level, is not ideal for deriving data for a high mountain area, especially with thousand meters differences in elevation.

(5) The number of plant species identified by sedaDNA was obviously much lower than that of current species pool in this study area. Despite of biological interaction and evolutionary adaption of alpine plant species, is the identified number of historical plant species large enough to predict future biodiversity change?

Authors' Response: We agree that our approach underestimates the total plant taxa richness, mainly because of limitations in the current taxonomic reference database and due to the limited specificity of the marker used. That said, we are confident that the sedaDNA approach is a powerful tool for tracking the long-term relative changes of taxa richness, as we found that our observed temporal richness pattern is not sensitive to the dataset used (please see the new Supplementary Table 1) and also because species-level richness is rather similar to richness at lower taxonomic levels (that can be resolved by the marker).

New comment: Total plant taxa includes various plant functional groups, such as tree, shrub, herb, etc. So, which group was highly underestimated and to what extent affected prediction of future biodiversity change? Again, it seems responses of different plant functional groups to future environmental changes are mixed. If the relationship between total plant taxa richness and alpine habitat in the time period of 10 – 0 ka was used for all plant functional groups, it means you assume that such relationship among plant functional groups is consistent, which obviously against the reality. For example, migration distances between trees and herbs are different, which determined the colonization process after glacier and snow retreat, hence showing different relationships with changes of alpine habitat extent.

The responses are in blue. The revisions are marked in red in the revised manuscript with Microsoft Word's Track Changes in bubbles. The comments were separated into several parts and responded to point by point.

Reviewers' comments:

Reviewer #2 (Remarks to the Author):

L. 372-373: uncertain what '0.25%' is referring to - is this minimum relative read abundance?

Response: It is referring to a maximum relative read abundance of 0.25% at least. We have indicated this. Please see lines 467-468.

The responses are in blue. The revisions are marked in red in the revised manuscript with Microsoft Word's Track Changes in bubbles. The comments were separated into several parts and responded to point by point.

Reviewers' comments:

Reviewer #3 (Remarks to the Author):

New comment (1): I doubt that the catchment in this study covers all microrefugia in alpine ecosystems. Microrefugia is actually a combination of several biotic and abiotic factors, which itself is a sophisticated term and hard to quantify. Anyway, a misunderstanding about SDMs should be avoided. I would not say SDMs ignore microrefugia. On the contrary, the SDMs consider hundreds or thousands sampling points which highly related to microhabitat information. Perhaps it is good to read some papers like 'Microclimate and demography interact to shape stable population dynamics across the range of an alpine plant', 'Extinction debt of high-mountain plants under twenty-first-century climate change'. Even if the catchment covered all microrefugia, one sediment core did not record any information relating to microrefugia, which is exactly the point that I argue you did not take microrefugia into account to predict future biodiversity change. I totally agree that the method used in this study is a new and interesting way to understand alpine biodiversity change. However, since the authors claimed in the abstract that their findings challenge the idea that future warming and treeline rise is expected to cause an increase of plant diversity in present alpine habitats, I think they should compare their prediction results to those from SDMs (as far as I know, there are also some predictions based on SDMs for the next century) and discuss advantages or limitations compared with SDMs. Otherwise, it would be very hard for audiences to understand the superiority of their method.

Response: Thank you for your comment. We understand the reviewer's concern that our wording raised the impression that we criticize SDM approaches. That was not our intention and we revised the text to provide a more balanced discussion of the available methods. We now clearly name the limitations of our simulation and refrain from using the term "prediction". Instead, we now use the phrase "by evidence from/by analogy to the past" when comparing our results to predictions from SDMs. Also, we have deleted all text related to a potentially misleading micro-macrorefugia debate. Instead, we indicate that results from our rather simple approach should be evaluated by more sophisticated approaches that include single taxa and consider migrational lags. Furthermore, we have adjusted the text in the abstract (lines 4-5 and 10-13). To summarize, we do not want to criticize other approaches and clearly state all limitations of our approach that you mentioned. We hope that the revised text meets your approval as the novelty is clearly on the side of plant diversity reconstruction and not on the side of prediction.

New text lines 4-5: “Hence, an increase in upper elevation diversity is expected in the course of warming-related treeline rise.”

Revised text lines 10-13: “Based on these inferred dependencies, our simulation yielded a substantive decrease in plant taxa richness in response to warming-related alpine habitat loss over the next centuries. Accordingly, efforts of Tibetan biodiversity conservation should include conclusions from palaeoecological evidence.”

New text lines 232-234: “An upward expansion of montane taxa and a loss of high alpine taxa in the study area agree with predictions from a comprehensive species distribution modelling approach for the Hengduan Mountains⁴.”

New text lines 285-289: “Our study indicates that time-series investigations from palaeoecological investigations using *sedaDNA* can inform decision-making in nature conservation by revealing potential plant responses to changing environments and, when used alongside modelling studies of modern species distributions, create a fuller picture of plant dynamics.”

4. Liang, Q. et al. Shifts in plant distributions in response to climate warming in a biodiversity hotspot, the Hengduan Mountains. *J. Biogeogr.* 45, 1334–1344 (2018).

New comment (2): It is not because the elevation range (4,400–4,800 m) makes Lake Naleng catchment a very typical lake to conduct such study, as a good sediment sample core actually matters. My argument is not how ‘typical’ the catchment is, which you could not quantify, but how one core information provides comprehensive information of vegetation in rugged terrain of vast Tibet plateau. For example, if there were two or three catchments separated by a mountain ridge, which is very common in alpine ecosystems, and you obtained some cores from the catchments, then the differences among the cores would tell how confidence the derived information is, especially at different time period.

Response: We thank the reviewer for this comment. If we understand correctly, your comment raises two questions. (1) Are *sedaDNA* analyses of one core representative of the Naleng catchment? (2) Is the Holocene vegetation change in Lake Naleng’s catchment representative of the Hengduan alpine area, given that we have no controls from other nearby lakes? We have answered these questions here and point to the (new) text in the manuscript.

- (1) Previous studies that have compared the sedaDNA results with the vegetation in the lake catchment found good relationships (Alsos et al., 2018, Niemeyer et al., 2017). This is confirmed by our sedaDNA record that tracks the expected invasion of *Picea* in the catchment during the early Holocene warming. Furthermore, sedimentation in a lake is generally considered spatially relatively homogenous, as indicated by many seismic studies and sediment replicate studies (e.g., Bird et al., 2014; Sun 2015). Because our sediment core does not show signs of disturbance and has a robust age-depth model, there is no reason for us to assume that the sediments are not representative of the lake in general. Thus, we are confident that our results reflect the catchment signal of past vegetation change.
- (2) We provided arguments that the major vegetation pattern in our studied catchment is representative of the Tibetan Plateau (lines 114-116). Of course, differences such as the lack of invasion of individual taxa, might still occur in Lake Naleng's catchment compared to other catchments covering the same elevational range. However, our study only targets the major patterns, for example doubling of plant richness. Our reasoning aligns with a modern study that found a non-random vegetation composition in the alpine belt of the Hengduan Mountains and identified environmental filtering as the main assembly process (Li et al., 2014) (see new text below).

While we are confident that the stated results represent major vegetation and plant biodiversity changes in Hengduan Mountain, further studies would be beneficial to support or refine our results. As lake-sediment-core analyses in general, and *sedaDNA* analyses in particular, are very time-consuming, a replicate study would not be affordable as part of this study, but we hope we or other groups can take up the challenge in the future. We believe that our findings provide strong motivation to do so.

New text lines 125-128: “This reasoning aligns with a modern study that indicates a non-random vegetation composition in the alpine belt of the Hengduan Mountains and identified phylogenetic clustering of alpine plant taxa in connection with environmental filtering²⁴”

24. Li, X. H., Zhu, X. X., Niu, Y. & Sun, H. *Phylogenetic clustering and overdispersion for alpine plants along elevational gradient in the Hengduan Mountains Region, southwest China: Phylogenetic structure along elevational gradient. J. Syst. Evol.* **52**, 280–288 (2014).

Bird, B. W. et al. *A Tibetan lake sediment record of Holocene Indian summer monsoon variability. Earth and Planetary Science Letters* **399**, 92–102 (2014).

Sun, W., Zhang, E., Jones, R. T., Liu, E. & Shen, J. Asian summer monsoon variability during the late glacial and Holocene inferred from the stable carbon isotope record of black carbon in the sediments of Muge Co, southeastern Tibetan Plateau, China. The Holocene 25, 1857–1868 (2015).

New comment (3): Treeline position highly affects the changes of alpine habitat extent defined in this study. However, if you can not prove that you obtained reliable dynamic changes of treeline, how confident the predictions of alpine habitat extent are?

Again, the relationship between total plant taxa richness and alpine habitat in the time period of 10–0 ka is too rough to predict future biodiversity change since some important processes are ignored, such as snow cover change, upward shift of low land plant species, species interaction... etc, which are widely believed as key processes affecting species richness in alpine ecosystems. Moreover, the prediction mixed the processes of trees and low stature plants above treeline, which also could result in unreliable predictions since responses of, for example, trees and herbs are obviously different.

Response: We appreciate the reviewer's skepticism that our predicted treeline changes are accurate enough to predict alpine habitat extent. We also agree that other processes, such as snow cover change and species interactions, may further influence habitat extent or plant taxa richness. We addressed these arguments in the revised version of the manuscript by an extended discussion: We present two main arguments that treeline changes are sensitive to temperature change in the region, at least on centennial to millennial time scales. (1) Pollen data from several sites (Fig. #3.1) agree with the pollen data and sedaDNA data from Lake Naleng, indicating that forests expanded into higher elevations under warming in the early- to mid-Holocene and retreated to lower elevations during the late Holocene cooling. (2) Modern observation indicates that warm temperatures on the southeastern Tibetan Plateau in the past 100 years³⁸ triggered upward moving treelines. Thus, our assumption that treelines will rise in the course of future warming is reasonable. However, the pace of treeline response to warming is less certain, as it is likely affected by various processes, including interspecific competition, forest-shrub interaction, dispersal variations and/or even extreme climate events. We think that differences in the response time of trees and low stature plants might be relevant on short time scales, but, as indicated by the observed consistent compositional change and significant treeline-richness relationship in our study, such differences are likely of minor importance over long time scales. In addition to the editor's suggestion, we toned down our inferences and clearly discuss the limitations of our simulation approach.

New text lines 235-244: “Our approach has several shortcomings. It assumes that tree-line change is sensitive to temperature change in the region. Although this assumption is supported by palaeoecological evidence showing that forests expanded into higher elevations under warming during the early- to mid-Holocene and retreated to lower elevations during the late Holocene cooling²², and by modern observations of an upward shifting treeline on the southeastern Tibetan Plateau in the past 100 years³⁷, the pace of treeline response is observed to lag the temperature warming in some mountain regions due to a variety of processes. Such processes include interspecific competition, forest-shrub interactions, dispersal variations or even extreme climate events^{37,38}. So, our temperature-treeline-richness relationship may therefore be correct on a millennial time scale but may overestimate changes on shorter time scales.”

Fig. #3.1 The location of fossil pollen records on the southeast Tibetan Plateau surrounding Lake Naleng (4200 m a.s.l.). (Lake Ren, 4450 m.a.s.l., Lake Yidun, 4470 m a.s.l., Lake Muge, 3780 m a.s.l., Lake Wuxu, 3705 m a.s.l., reviewed by Ref. 22)

22. Chen, F. et al. *Climate change, vegetation history, and landscape responses on the Tibetan Plateau during the Holocene: A comprehensive review. Quat. Sci. Rev.* 243, 106444 (2020).

37. Liang, E. et al. *Species interactions slow warming-induced upward shifts of treelines on the Tibetan Plateau. Proc. Natl. Acad. Sci.* 113, 4380–4385 (2016).

38. Alexander, J. M. et al. *Lags in the response of mountain plant communities to climate change. Glob. Change Biol.* 24, 563–579 (2018).

New comment (4): Temperature lapse rate might be a good solution if there’s no available adjacent climate station. However, climate data from Chengdu station, a big city with high

urbanization level, is not ideal for deriving data for a high mountain area, especially with thousand meters differences in elevation.

Response: We apologize if our explanation was misleading. We actually did not use data from the Chengdu climate station alone. Instead, we used an integrated temperature lapse rate in the Hengduan Mountains that was calculated by Li et al. (2013) based on multiple climate stations, as Fig. #3.2 indicates (region 18, ref. 64), including stations from areas adjacent to our study site. We have added further information in lines 415-416 and line 424-425.

Fig. #3.2 Distribution of the 754 Chinese national meteorological stations. Stations with '+' maker are used for lapse rate calculation and stations with open circle are used for validation. The rainbow scale bar depicts elevation (m) from SRTM digital elevation model. (Figure resource: ref. 64).

64. Li, X. et al. Near-surface air temperature lapse rates in the mainland China during 1962-2011. *J. Geophys. Res. Atmospheres* **118**, 7505–7515 (2013).

New comment (5): Total plant taxa includes various plant functional groups, such as tree, shrub, herb, etc. So, which group was highly underestimated and to what extent affected prediction of future biodiversity change? Again, it seems responses of different plant functional groups to future environmental changes are mixed. If the relationship between total plant taxa richness and alpine habitat in the time period of 10–0 ka was used for all plant functional groups, it means you assume that such relationship among plant functional groups is consistent, which obviously against the reality. For example, migration distances between trees and herbs are different, which determined the colonization process after glacier and snow retreat, hence showing different relationships with changes of alpine habitat extent.

Response: Thank you for this comment. Indeed, we did not investigate functional groups separately; however, we investigated single families which often comprise taxa from similar

functional groups. We expanded the discussion by pointing out that our approach considers richness in whole or species-rich alpine families.

Generally, plant taxa richness may be underestimated due to the lack of a site-specific taxonomic reference database, as we mentioned in lines 143-146. We have no specific evidence that the systematic underestimation of richness in specific functional groups could bias future predictions due to their underrepresentation in the reference database. We, therefore, added this as a potential shortcoming in our evaluation. Overall, we assume that we may underestimate richness in all functional groups to some extent because all functional groups are represented with reasonable share. The *Picea* and *Abies* DNA were detected, comparable to modern montane forests within the lake's catchment, consisting of conifers (*Abies aquamata*, *A. faxonia*, *Picea likiangensis*, and *P. purpurea*). The Salicaceae, *Rhododendron*/Ericaceae, and Cupressaceae DNA tracks the modern alpine shrubs belt (e.g., *Salix vaccinoides*, *Rhododendron telmateium*, and *Juniperus*) within the lake's catchment. Species-rich alpine herbs (e.g., Asteraceae, Orobanchaceae, Ranunculaceae, and Saxifragaceae) on the Tibetan Plateau are also rich in our *sedaDNA* data. Therefore, we consider that the *sedaDNA*-based plant taxa richness can be regarded as a semi-quantitative proxy of taxa richness in our study.

New text lines 244-248: "Furthermore, our approach considers plant richness as a whole or focus on certain alpine families. Therefore, the habitat gain and loss of individual taxa or specific functional groups cannot be evaluated. Hence, our simulated taxa loss in relation to shrinking alpine habitat extent should be treated as a potential pattern by analogy to the past."

Revised text lines 248-249: "It requires confirmation from a more sophisticated species-specific approach that also considers realistic migrational lags."

Reviewers' Comments:

Reviewer #3:

Remarks to the Author:

I believe the authors have made sufficient revisions to my concerns. I am happy to see that the authors well discussed the limitations of their approaches, which I believe is important for ecologists to understand the underlying mechanisms in this study area. I thus congratulate to authors for their great findings.

The responses are in blue.

REVIEWERS' COMMENTS

Reviewer #3 (Remarks to the Author):

I believe the authors have made sufficient revisions to my concerns. I am happy to see that the authors well discussed the limitations of their approaches, which I believe is important for ecologists to understand the underlying mechanisms in this study area. I thus congratulate to authors for their great findings.

Response: We sincerely appreciate you for reviewing our manuscript.